Resource

# Mechanisms of NLRP3 activation and inhibition elucidated by functional analysis of disease-associated variants

The NLRP3 inflammasome is a multiprotein complex that mediates caspase-1 activation and the release of proinflammatory cytokines, including interleukin (IL)-1β and IL-18. Gain-of-function variants in the gene encoding NLRP3 (also called cryopyrin) lead to constitutive inflammasome activation and excessive IL-1β production in cryopyrin-associated periodic syndromes (CAPS). Here we present functional screening and automated analysis of 534 NLRP3 variants from the international INFEVERS registry and the ClinVar database. This resource captures the effect of NLRP3 variants on ASC speck formation spontaneously, at low temperature, after inflammasome stimulation and with the specific NLRP3 inhibitor MCC950. Most notably, our analysis facilitated the updated classification of NLRP3 variants in INFEVERS. Structural analysis suggested multiple mechanisms by which CAPS variants activate NLRP3, including enhanced ATP binding, stabilizing the active NLRP3 conformation, destabilizing the inactive NLRP3 complex and promoting oligomerization of the pyrin domain. Furthermore, we identified pathogenic variants that can hypersensitize the activation of NLRP3 in response to nigericin and cold temperature exposure. We also found that most CAPS-related NLRP3 variants can be inhibited by MCC950; however, NLRP3 variants with changes to proline affecting helices near the inhibitor binding site are resistant to MCC950, as are variants in the pyrin domain, which likely trigger activation directly with the pyrin domain of ASC. Our findings could help stratify the CAPS population for NLRP3 inhibitor clinical trials and our automated methodologies can be implemented for molecules with a different mechanism of activation and in laboratories worldwide that are interested in adding new functionally validated NLRP3 variants to the resource. Overall, our study provides improved diagnosis for patients with CAPS, mechanistic insight into the activation of NLRP3 and stratification of patients for the future application of targeted therapeutics.

✉e-mail: seth.masters@hudson.org.au

Inflammasomes are multimeric protein complexes that play an important role in innate immune signaling[1,2]. NLR family pyrin domain-containing 3 (NLRP3), also known as cryopyrin, is one of the most broadly studied inflammasome sensors[2]. NLRP3 is composed of an N-terminal pyrin domain (PYD), a central NACHT domain and a C-terminal leucine-rich repeats (LRR) domain[3]. The NACHT domain of NLRP3 mediates ATP hydrolysis, which is important for switching the conformation of NLRP3 from the active to the inactive form[3–5]. In the inactive, ADP-bound state, human NLRP3 forms a decamer via LRR–LRR interactions[5]. Activation of NLRP3 requires two steps, including priming and activation[2]. Priming usually involves nuclear factor (NF)-κB signaling downstream of Toll-like receptors or another signal that leads to increased protein synthesis of NLRP3 and pro-IL-1β and post-translational modifications (PTMs) of NLRP3 such as deubiquitination, phosphorylation, sumoylation and palmitoylation[6]. NLRP3 can then be activated by a wide range of triggers, such as the bacterial toxin nigericin or danger-associated molecular patterns such as ATP and particulates such as uric acid crystals[7,8]. Many NLRP3 activators converge to induce K⁺ efflux, except for the TLR7 ligand, imiquimod, which triggers the production of mitochondrial reactive oxygen species (ROS)[9,10]. In the active, ATP-bound state, the PYD domain of NLRP3 oligomerizes and recruits the adaptor protein apoptosis-associated speck-like protein containing a caspase recruitment domain (ASC) via homotypic PYD–PYD interactions[11]. Another essential component of the active NLRP3 inflammasome disk is the centrosomal NIMA-related kinase 7 (NEK7)[11,12]. NEK7 facilitates NLRP3 activation by interacting with NLRP3 at multiple surfaces in the LRR and NACHT domains[12]. The activation of caspase-1 within the inflammasome complex drives inflammatory signaling by cleavage of pro-IL-1β and pro-IL-18, which are key mediators in autoinflammatory diseases[2,13,14]. Caspase-1 also cleaves the pore-forming protein, gasdermin D, which mediates pyroptosis and the release of IL-1β and IL-18 (ref. 2).

Gain-of-function variants in *NLRP3* promote excessive inflammasome activation and IL-1β overproduction and drive pathology in CAPS[14,15]. CAPS is a spectrum of diseases, divided into three clinical phenotypes with increasing severity called familial cold autoinflammatory syndrome (FCAS), Muckle–Wells syndrome (MWS) and neonatal-onset multisystem inflammatory disease (NOMID)[14]. Currently, multiple anti-IL-1 treatments, including anakinra, canakinumab and rilonacept are registered and effective for patients with CAPS[13,14]. Early administration of treatments could improve quality of life and avoid organ damage[14]. For example, early treatment of MWS can reverse or halt the progression of hearing loss which can be irreversible if treatment is delayed[16,17]. Moreover, patients with low-penetrance variants or somatic mosaicism in NLRP3 might present with atypical phenotypes, leading to delayed diagnosis and treatment[14]. Therefore, the classification of NLRP3 variants for their pathogenicity can support early diagnosis and treatment of CAPS, avoiding potential organ damage and improving quality of life[14].

Genetic variants in *NLRP3* are frequently classified as pathogenic, likely pathogenic, benign, likely benign or as a variant of uncertain significance (VUS). These classifications are based on population data, inheritance, computational predictions of pathogenicity and functional data[18]; however, functional analyses, which provide strong evidence for variant classification, are missing for 83% of variants in the international registry of autoinflammatory disease variation, INFEVERS[19–22] and for most variants from the ClinVar database[23,24].

New treatment approaches targeting NLRP3 could also be beneficial for patients with CAPS. For instance, MCC950, a small molecule inhibitor for NLRP3 has been shown to inhibit ATP hydrolysis by binding to and stabilizing the NACHT domain[5,25–27]. Additionally, MCC950 can inhibit IL-1β production in peripheral blood mononuclear cells (PBMCs) of patients carrying different NLRP3 variants[28–30]; however, MCC950 failed to inhibit inflammasome activation in macrophages derived from transgenic mice carrying the L355P NLRP3 variant[31], and in human

monocytes simultaneously expressing E527K and D648Y variants[30]. These results show that different NLRP3 variants have discrepancies in responding to MCC950-mediated inhibition. A comprehensive screening for the response of all NLRP3 variants to inhibitors, such as MCC950, could help anticipate the efficacy of treatments targeting NLRP3.

To fill the knowledge gap surrounding NLRP3 variants and their pathogenicity, we developed a functional screening assay and automated analysis pipeline using variants from the INFEVERS registry[19–22] and the ClinVar databases[23,24]. By assessing the activity of NLRP3 variants in ASC speck formation, we proposed a new functional score named ASC$_{50}$ to rank and re-evaluate the pathogenicity classification. These results prompt reconsideration for pathogenicity classification for nearly 10% of variants in the INFEVERS registry, where affected patients will now be more likely to receive appropriate therapy and counseling.

## Results

### Functional testing of NLRP3 activity reveals pathogenic CAPS variants

To comprehensively understand the activity of CAPS-associated NLRP3 variants, we developed an automated functional test that can be applied at scale for all known coding variants in NLRP3. From an initial 264 variants present on INFEVERS at the time of access, we filtered out non-coding and synonymous variants, leaving 208 missense NLRP3 variants for screening (Extended Data Fig. 1a and Supplementary Table 1)[19–22]. We included an additional 313 NLRP3 variants from the ClinVar database[23,24] and 13 from personal contacts (Supplementary Table 1). Given that the ASC speck formation is a hallmark of inflammasome activation[3,32,33], we used this as a readout for our screening assay. We confirmed the ASC speck formation using image-based flow cytometry (Extended Data Fig. 1b,c)[34]. As expected, we found that cells with a higher level of NLRP3 expression had a higher number of ASC specks (Extended Data Fig. 1d,e). Therefore, we used the expression level of wild-type (WT) NLRP3 (tagged with GFP), which generates 50% ASC specks to define the baseline. Gain-of-function variants will generate 50% ASC specks at much lower levels of NLRP3 expression. Specifically, we developed an R-based program for automated gating and analysis of ASC speck percentage at different levels of NLRP3 expression (Extended Data Fig. 1f; https://github.com/Seth-Masters-Lab/Speck-Assay-Automated-Workflow.git). We then calculated the amount of NLRP3 that is required to form ASC specks in 50% of the population (EC$_{50}$) and compared the EC$_{50}$ values of NLRP3 variants to WT, which generates an EC$_{50}$ ratio. We defined ASC$_{50}$ as the relative difference of EC$_{50}$ between the variant and WT, and positive ASC$_{50}$ corresponds to NLRP3 hyperactivity. We used this method to analyze the basal activity of all 534 variants for NLRP3 found in INFEVERS, ClinVar databases and from clinician contacts (Fig. 1a). Moreover, we show that the ASC$_{50}$ value does not correlate with NLRP3 expression levels, confirming the overall data representativity (Extended Data Fig. 2a). We also confirmed that NLRP3, ASC, IL-1β and IL-18 are not expressed in HEK293T cells unlike in myeloid cells, whereas NEK7 is expressed in both HEK293T cells and myeloid cells tested (Extended Data Fig. 2b).

Variants that lead to NLRP3 hyperactivity were concentrated in the NACHT domain, specifically subdomains NBD and HD2, calculated based on the number of strongly active variants relative to the length of a domain (Fig. 1b). Hyperactive variants were also found in the PYD and LRR domains; however, with low frequency (Fig. 1b). To investigate whether ASC$_{50}$ correlates with CAPS disease phenotype, we compared variants leading to NOMID, MWS or FCAS and variants with overlapping symptoms (Fig. 1c). Indeed, we found that NOMID variants showed stronger activation, whereas MWS and FCAS variants had lower NLRP3 activation levels. Given the complicated genetic status of each variant, we compared the ASC$_{50}$ for pathogenic and likely pathogenic variants occurring as germline, mosaic or both (Extended Data Fig. 2c). We found that variants found only as mosaic are more

active than those found as the germline. Additionally, we compared variants that occurred at the same site but with different amino acid changes leading to either NOMID, MWS, FCAS or overlapping syndromes (Fig. 1d and Extended Data Fig. 2d,e)[35–37]. We observed a trend in the activation status of the variants, which agrees with the disease presentation. To confirm whether this trend is also present in myeloid cells, we knocked out NLRP3 in human THP-1 cells and overexpressed GFP-tagged WT or NLRP3 variants using a doxycycline-inducible lentiviral vector (Extended Data Fig. 3a–d). This confirmed that the NOMID variant R262P had higher inflammasome activity as indicated by increased level of IL-1β and IL-18 release compared with that for MWS variant R262L and MWS/FCAS variant R262W (Extended Data Fig. 3d); however, given that patients carrying the same variant can still present with different manifestations[36–40], it should be noted that a functional assay would not replace careful clinical assessment.

As our assay has been broadly validated for the phenotype of patients with pathogenic variants, we sought to improve the classification of NLRP3 variants. First, we interrogated VUS for which the $ASC_{50}$ value was higher than 0.281, the lower bound of the 95% confidence interval for $ASC_{50}$ of symptomatic NLRP3 variants. The s.e.m. for $ASC_{50}$ was also considered for variants near the border of the cutoff. Together with literature review, database analysis and discussion with clinicians, 19 VUS from the INFEVERS registry and a further 20 VUS from the ClinVar database were ultimately recommended to be reconsidered for classification as likely pathogenic (Fig. 1e). This represented cases where disease presentation was considered atypical at the time of diagnosis, where inheritance analysis was not possible or annotation was incorrect. Second, we examined pathogenic or likely pathogenic variants where the $ASC_{50}$ value was below 0.281 in case they should be reclassified as likely benign or VUS (Fig. 1e). This could be achieved in five cases from INFEVERS and three from ClinVar, particularly associated with variants that are present in databases of nominally healthy individuals (Supplementary Table 1). Notably, we followed up on one patient with the novel de novo NLRP3 variant R605G, which was initially reported to be likely pathogenic[41]. Despite the persuasive genetic evidence and CAPS-like phenotype at diagnosis, follow-up reveals that symptoms have now largely resolved without substantial intervention, and the clearly negative result in our functional assay permitted reclassification as likely benign (Fig. 1e). Furthermore, we reconstituted these likely benign variants and VUS including R172S, Q225P, R556*, R605G and Q705K into NLRP3-deficient THP-1 cells (Extended Data Fig. 4a). We confirmed that their inflammasome activity as triggered by the NLRP3 activator, nigericin and measured by the level of IL-1β and IL-18, could not surpass WT. We also used PMA-differentiated THP-1 cells and tested the monosodium urate (MSU) crystals, K+ independent NLRP3 activator, imiquimod and another NLRP3 activator, LLOMe, which activates NLRP3 by inducing lysosomal rupture (Extended Data Fig. 4b)[8]. Both R605G and Q705K did not have increased inflammasome activity compared with WT. Overall, these results demonstrate that automated functional validation of NLRP3 is an efficient way to improve the diagnosis of CAPS.

### Loss-of-function NLRP3 variants are identified in the screening

We also identified NLRP3 variants with much lower $ASC_{50}$ value than WT during the screen in HEK293T cells (Extended Data Fig. 5a). Specifically, we observed a complete loss of ability to form ASC speck for NLRP3 variants, including R7C, R7H, D31V, Q45*, T46R, N62S, M70T and I74N (Extended Data Fig. 5b). Except for M70T, all these variants also failed to respond to stimulation with nigericin. All these loss-of-function variants occurred in the PYD domain of NLRP3 and could directly impact the ability to form filaments and interact with ASC. Indeed, a previous study tested recombinant NLRP3 PYD domain carrying a D31V variant and found it lost the ability to oligomerize into ordered polymers, which is essential for ASC interaction[42]. Additionally, Q225P and R556* whose $ASC_{50}$ values are also below 0, had no inflammasome activity

when expressed in THP-1 cells despite their ability to form ASC speck in HEK293T cells (Extended Data Figs. 4a and 5b). The health impact of these loss-of-function NLRP3 variants remains unknown; however, none appears as homozygotes in large databases such as gnomAD[43].

### Pathogenic NLRP3 variants drive structural disruptions for hyperactivity

The NACHT domain of NLRP3 hydrolyzes ATP into ADP, and coordinates the active NLRP3 conformation at the ATP-bound stage[3,4]. To understand how gain-of-function NLRP3 variants lead to its hyperactivity, we investigated variants that can affect the structural rearrangements that occurs during NLRP3 activation. We found that variants located at critical points mediating NLRP3 structural rearrangements all led to increased NLRP3 activity (Supplementary Table 1), consistent with previous study[5]. For example, we observed that the distance between F304 and C261 can be reduced during NLRP3 transformation from inactive to active status (Fig. 2a,b). When the phenylalanine was mutated into cysteine at 304, it had an increased chance of forming a disulfide bond with the other cysteine at 261 due to closer contact in the active NLRP3 conformation. Therefore, F304C may lead to the formation of a strong covalent bond locking the NLRP3 structure in the active status and thus promote NLRP3 activity.

We further investigated whether any of the validated variants may impact ATP binding thus leading to NLRP3 hyperactivity. In the NACHT domain of NLRP3, the nucleotide-binding subdomain contains a Walker A motif characterized by ATPase-specific P-loop and a Walker B motif which is a $Mg^{2+}$ cation-binding site[3,4]. NLRP3 variants, including G303D, F304L/C, D305N/G/H/A, E306K/D are all located in the Walker B motif and showed increased NLRP3 activity (Supplementary Table 1). Among them, D305H and E306K both change the negatively charged side chains into positive and promote the electrostatic interaction between NLRP3 variants and ATP (Fig. 2c). CAPS-associated variant sites outside the Walker motifs, including L413 and W416, are also in the direct contact interface with ATP. Both L413F and W416L could affect the ATP hydrolysis activity in a way that favors the NLRP3 active conformation. We also compared the $ASC_{50}$ for variants that occurred near the ATP-binding pocket, all of which had increased levels of NLRP3 activity (Fig. 2d).

We then investigated whether NLRP3 variants can promote active NLRP3 oligomerization or destabilize the inactive NLRP3 complex (Fig. 2e,f). Variants including W416L, F445L, A441P and K437N were at the interface between active NLRP3 oligomers and are in close proximity to V512 and E511 from the adjacent NLRP3 monomer (Fig. 2e). These variants might promote the formation of active NLRP3 oligomers. In the inactive NLRP3 oligomer structure, E690K and E692K can form multiple hydrogen bonds with amino acids in the LRR domain and thus move the acidic loop away from the LRR of the adjacent monomer (Fig. 2f). These variants might destabilize the inactive NLRP3 complex. We and others also showed two variants in the PYD domain that led to hyperactive NLRP3 (Fig. 2g)[42]. The variant H51R alters the side chain orientation, which could promote E15 hydrogen bond formation with K24 from the adjacent PYD and thus, NLRP3 hyperactivity. D21H at the other interface of the PYD filament, seems to form a hydrogen bond with D60 from the adjacent PYD domain and thus promotes PYD–PYD interactions. Overall, the validated pathogenic variants in NLRP3 could use distinct mechanisms to promote NLRP3 activation (Extended Data Fig. 5c).

### NLRP3 variants that are hypersensitive to stimulation

Given that many CAPS-associated NLRP3 VUS had similar or only slightly increased $ASC_{50}$ when compared with WT (Fig. 1a and Supplementary Table 1), it is possible that they could be hyperresponsive to NLRP3 activating triggers. Therefore, we tested the NLRP3 activator, nigericin, which mediates K+ efflux and triggers the formation of ASC speck formation with NLRP3 (Fig. 3a and Extended Data Fig. 6a,b)[8]. We found that NLRP3 variants such as Y861H and R920Q had increased activity in

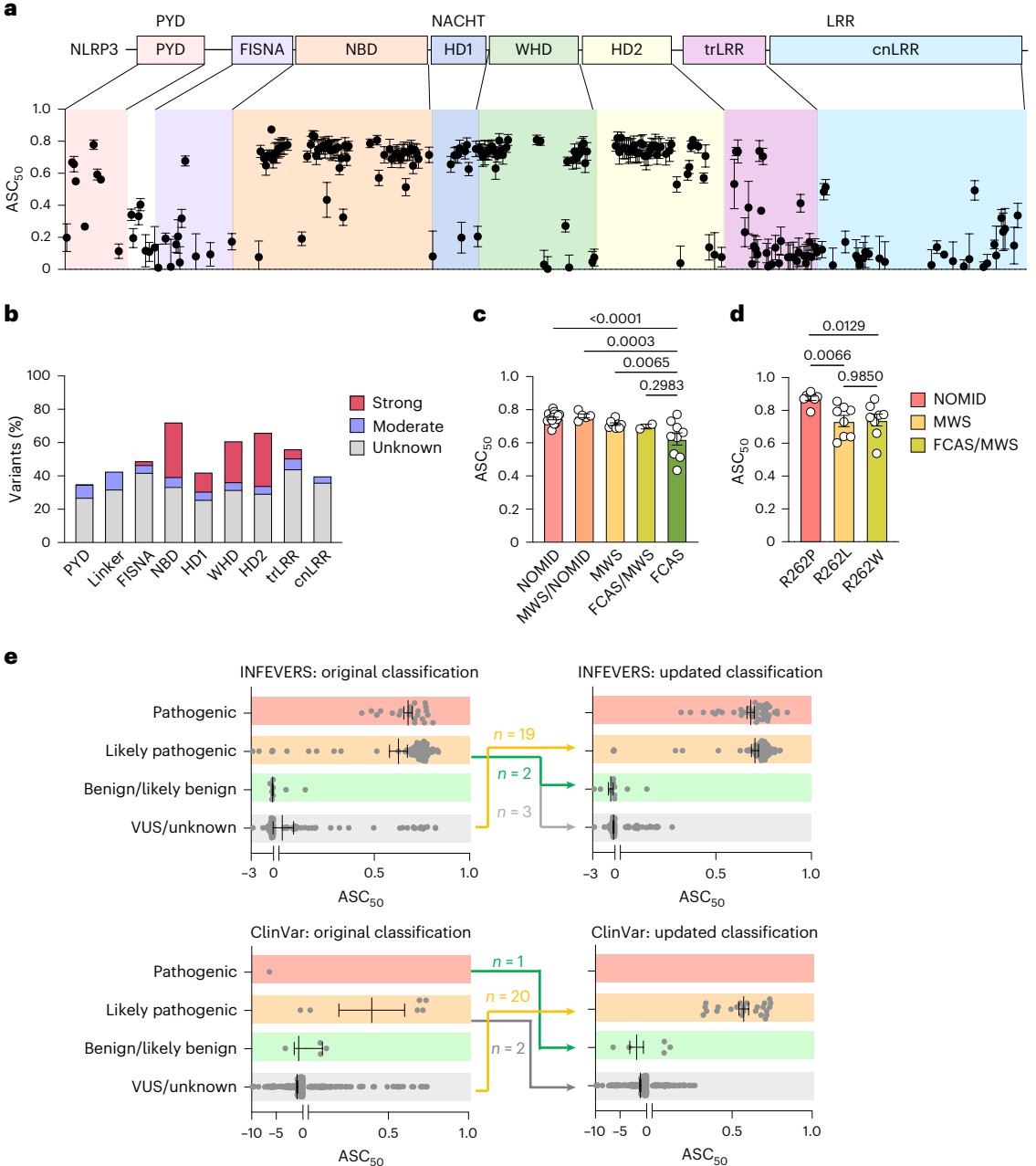

**Fig. 1 | ASC₅₀ of NLRP3 variants is positively correlated with pathogenicity.**
**a**, NLRP3 variants with $ASC_{50}$ between 0 and 1. ASC-BFP HEK392T cells were analyzed using flow cytometry 12 h post-transfection (bottom). The location of each variant is color-coded as in the schematic diagram of NLRP3 domain organization (above). PYD, pyrin domain; FISNA, fish-specific NACHT-associated domain; NBD, nucleotide-binding domain; HD, helical domain; WHD, winged helix domain; trLRR, transition leucine-rich repeat domain; cnLRR, canonical LRR domain. **b**, Number of strongly active, moderately active or unknown significance variants relative to the length of each domain or subdomain. **c**, $ASC_{50}$

of INEFVERS variants identified in patients with NOMID, MWS or FCAS or patients with overlapping symptoms. **d**, $ASC_{50}$ of variants at R262 found in NOMID, MWS or patients with FCAS/MWS. **e**, $ASC_{50}$ of 206 NLRP3 variants from INFEVERS (top) and 307 variants from ClinVar (bottom) grouped on classification before (left) and after (right) update. Each dot represents a single variant (**a**,**c**,**e**) or one independent repeat (**d**). Data were pooled from 3–12 independent repeats (n value for each variant is listed in Supplementary Table 1,a,c–e, mean and s.e.m. in **a**,**c**–**e**). One-way analysis of variance (ANOVA) with Dunnett's multiple-comparisons test in **c**,**d**).

response to nigericin treatment compared with WT (Fig. 3a and Supplementary Table 2). Overall, the variants that sensitize NLRP3 for K⁺ efflux were mostly located on the surface of NLRP3 (Fig. 3b). We confirmed the increased activity of Y861H and R920Q in response to nigericin compared with WT NLRP3 in THP-1 cells (Fig. 3c,d and Extended Data Fig. 6c,d). Moreover, several NLRP3 variants that had reduced activity, including the premature truncation R556*, can also respond to nigericin treatment in HEK293T cells but not in THP-1, suggesting that these variants retained the ability to interact with ASC in HEK293T cells

(Extended Data Figs. 4a and 5b). Other examples include G328A, R605G, P352L, F410S, R699*, D498V and L917R (Extended Data Fig. 6a,b,e). We further tested whether the activity of these nigericin-hyperresponsive NLRP3 variants can be inhibited by blocking K⁺ efflux. Indeed, increasing concentrations of KCl in the medium did not suppress the basal activity of NLRP3 or the autoactivated NLRP3 variants but did inhibit nigericin-induced activation (Extended Data Fig. 6f,g).

The variants that sensitize NLRP3 for K⁺ efflux are positioned in close contact with NEK7 (Extended Data Fig. 6h)[12,44]. To investigate

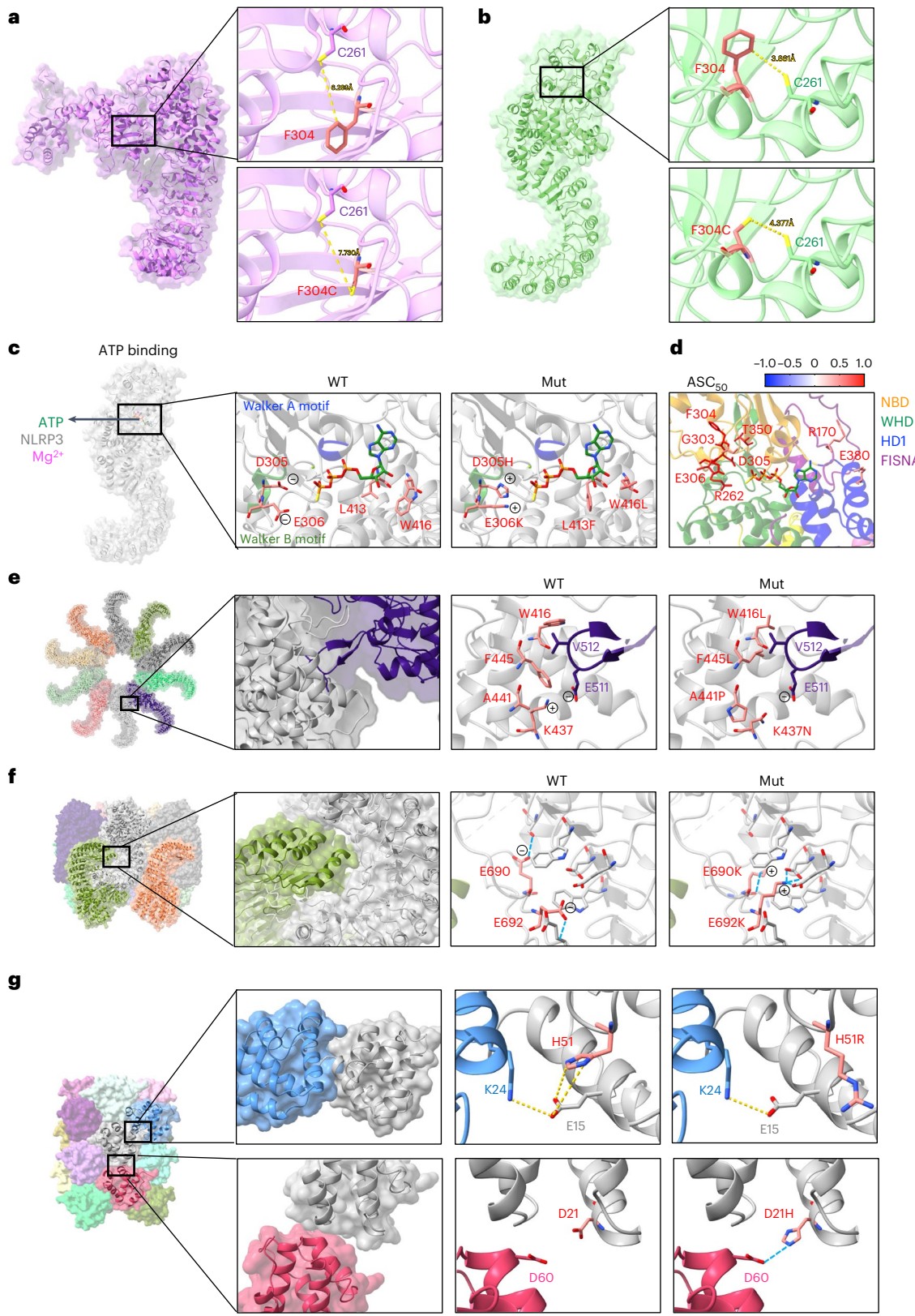

**Fig. 2 | Structural analysis of NLRP3 variants. a,b.** Distance between Cys261 and Phe304 or Cys261 in the inactive (**a**) or active (**b**) NLRP3 structure (PDB: 7PZC and 8EJ4). **c**, Molecular interactions of ATPγS with WT or mutant (Mut) NLRP3 (PDB: 8EJ4). **d**, Mapping of variants within contact range with ATPγS in active NLRP3 structure (PDB: 8EJ4). Residues are highlighted as sticks and in color scale based on the mean value of $ASC_{50}$ of variants that occurred at the designated sites (*n* value for each variant is listed in Supplementary Table 1). **e**, Variants at the interface of active NLRP3 oligomers (PDB: 8EJ4). **f**, Variants at the interface of inactive NLRP3 decamer (PDB: 7PZC). **g**, Variants at the interface of PYD filament (PDB: 7PZD).

whether the nigericin-sensitive variants play a role in mediating the interaction with NEK7, we overlayed the structures of active and inactive NLRP3 and compared the location of variants relative to NEK7 (Extended Data Fig. 6h). We observed that the acidic loop in the LRR of inactive NLRP3 structure could clash with NEK7 binding to inactive NLRP3, suggesting that the acidic loop could be repositioned during NLRP3 activation. Previous studies had shown that inhibition of $K^+$ efflux reduced NLRP3 binding with NEK7 (ref. [45]), so we hypothesized that the nigericin-hypersensitive NLRP3 variants may show hyperactivity with an increased level of binding with NEK7. To address this hypothesis, we overexpressed nigericin-hyperresponsive NLRP3 variants in HEK293T cells expressing both ASC-BFP and NEK7[K64M]-mCherry and analyzed $ASC_{50}$ (Extended Data Fig. 6i). Given that overexpressing WT NEK7 in cells for a long term can cause cell cycle arrest[46], we used the kinase-inactive mutant NEK7[K64M], which had a similar level of interaction with NLRP3 as the WT[44]. We used NLRP3 variants, including G757R and G757A, which have been shown to have increased interaction with NEK7 as positive controls[44]. We found that overexpressing NEK7[K64M] increased the activity of G757R and G757A, but not the nigericin-hyperresponsive NLRP3 variants. Additionally, we knocked out NEK7, or NEK7 together with NLRP3, from THP-1 cells to create a system in which to check whether NEK7 is important for the increased activity of reconstituted NLRP3 variants (Extended Data Fig. 7a,b). We found that the activity of G757R was reduced in the absence of NEK7 (Extended Data Fig. 7c,d); however, Y861H and R920Q remained hyperresponsive to nigericin stimulation in THP-1 cells even in the absence of NEK7 (Extended Data Fig. 7e–h), confirming that NEK7 does not contribute to the hyperactivity of some nigericin-sensitive NLRP3 variants.

To look for another mechanism by which Y861H and R920Q may sensitize NLRP3 to stimulation, we examined LRR–LRR interfaces in the structure of the inactive NLRP3 decamer (Fig. 3e). We found that nigericin-sensitizing variants, especially R920Q and Y861H could directly impact the positioning of the acidic loop and thus disrupt the inactive NLRP3 complex formation. Specifically, R920 coordinates the positioning of the acidic loop by attracting D702 and repelling H700; however, a variant from arginine into glutamine might lead to a stronger interaction with D702 and H700, pulling the acidic loop toward the concave side of LRR. Furthermore, we observed that by changing the tyrosine at 861 changed into a histidine, a hydrogen bond can be formed with E693. Therefore, we suggest that the positively charged acidic loop of NLRP3 might undergo conformational changes when the ionic environment is altered in response to nigericin, leading to increased sensitivity for the specific variants identified in this region.

**Cold exposure triggers variant-specific NLRP3 hyperactivation**

NLRP3 variants including L355P and D305N, have been shown to be activated by cold temperature exposure[47]. To investigate whether the CAPS-associated NLRP3 variants have a general feature of cold responsiveness, we screened a subset of NLRP3 variants which had similar activity compared with WT and those reported in patients with FCAS or CAPS-undefined (Supplementary Table 1 and Supplementary Table 3). We incubated the cells for 12 h at either 37 °C or 32 °C following transfection with NLRP3 variant-expressing plasmids and analyzed ASC speck formation using flow cytometry. Consistent with a previous finding[47], L355P and D305N led to increased NLRP3 activity after cold exposure, although these variant was already hyperactive when compared with WT at 37 °C (Fig. 4a and Supplementary Table 3). Other INFEVERS variants, including E629G, C261W, G303D, R262W, R262L and Y565N had increased NLRP3 activity when exposed to cold temperature. Among them, L355P, E629G, C261W, G303D and Y565N have all been identified from patients with FCAS[20,37,48,49], and R262W has been identified from both patients with FCAS and MWS[36,40]. Therefore, our results are consistent with the clinical manifestations presented in patients with FCAS. We also tested less-active ClinVar variants at reduced temperature and found that V643M, F257fs, T966fs, T440del, M661R and C633W

had significantly increased $ASC_{50}$; however, ClinVar does not indicate whether these are associated with FCAS (Extended Data Fig. 8a and Supplementary Table 3). The cold temperature-sensitive variants identified are mostly localized in the NACHT domain of NLRP3 (Extended Data Fig. 8b). We then asked whether this variant-led cold responsiveness of NLRP3 occurs at a specific site of the protein structure or requires specific amino acid changes. To answer this question, we compared the $ASC_{50}$ at 37 °C or 32 °C for different variants that occurred at the same site. For example, Y565N demonstrated increased $ASC_{50}$ at lower temperature, whereas Y565C showed no significant increase (Extended Data Fig. 8c). This was also true at positions R262 and E629, where both temperature-sensitive and -insensitive variants are found (Extended Data Fig. 8d,e). Finally, we found one profoundly cold-sensitive variant, L413V, whereas a different variant at this same site, L413F, was not cold sensitive (Fig. 4b,c). Moreover, the hyperactivity of L413V was not due to increased protein expression level at 32 °C (Extended Data Fig. 8f). We further confirmed in reconstituted THP-1 cells that L413V, but not L413M, had increased inflammasome activity in response to cold temperature, with a similar trend as in the FCAS variant, L355P (Extended Data Fig. 9a). Together, this argues that the specific location of a variant does not dictate temperature sensitivity, but that the precise quality of the amino acid change can potentiate cold responsiveness of NLRP3.

Given that we found that only one variant was profoundly potentiated at low temperature, we looked at it more closely. L413 is located at the hydrophobic region of the ATP-binding pocket, where variants such as L413F and L413M might enhance ATP binding and promote NLRP3 activity independent of temperature (Fig. 4d). Indeed, both phenylalanine and methionine have the potential to form stronger hydrophobic interactions with the ribose of ATP leading to NLRP3 hyperactivity, but this potential is not obvious with L413V (Fig. 4d,f). To understand how L413V is different from L413F and L413M, we used DynaMut to predict the energy changes in NLRP3 variants[50]. DynaMut is an algorithm that can estimate vibrational entropy, which is a measure of protein flexibility determined by the frequencies of normal vibrations in the protein, which relates to the space available for molecules to move within[50]. We found that L413V had higher vibrational energy and destabilized the NLRP3 structure in the active state (Fig. 4g). L413F and L413M promoted active NLRP3 stability with both having increased NLRP3 activity regardless of temperature (Fig. 4f,g). It is possible that in a low free-energy environment such as under cold temperature, the NLRP3 structure becomes more stable, and thus the effect of L413V on its activity can be detected. It could be the case that other variants that are also temperature-sensitive, but to a lesser degree than L413V, share a similar mechanism of action, but with reduced effect, by virtue of occurring more distal to the ATP-binding pocket.

**Post-translational modifications contribute to NLRP3 activity**

PTMs such as phosphorylation or ubiquitination can modulate NLRP3 inflammasome activity (Supplementary Table 5)[51,52]. Reported variants at S198, K567 and Y861 occur at sites of NLRP3 PTM, which might not be captured in the HEK293T cell line assay. Therefore, we reconstituted S198N, K567E, Y861H and Y861C into NLRP3-deficient THP-1 cells (Extended Data Fig. 9b–d). Phosphorylation of NLRP3 at S198 by mitogen-activated protein kinase 8 (MAPK8; also known as JNK1) is a key priming event that facilitates NLRP3 activation[53]. Indeed, phosphorylation-resistant S198N produced less IL-1β and IL-18 in response to nigericin when compared with similarly expressed WT NLRP3 (Extended Data Fig. 9b); however, MAPK8 is also active in HEK293T cells[54], as shown by the consistent result for S198N when compared with WT in our original screen (Supplementary Table 1). We also saw consistently increased activity for K567E in both HEK293T cells and THP-1 cells (Supplementary Table 1 and Extended Data Fig. 9c). This is in contrast to the published literature where mouse NLRP3 activity was promoted by ubiquitination at K565 (ref. [55]); however, that result was not confirmed with human NLRP3.

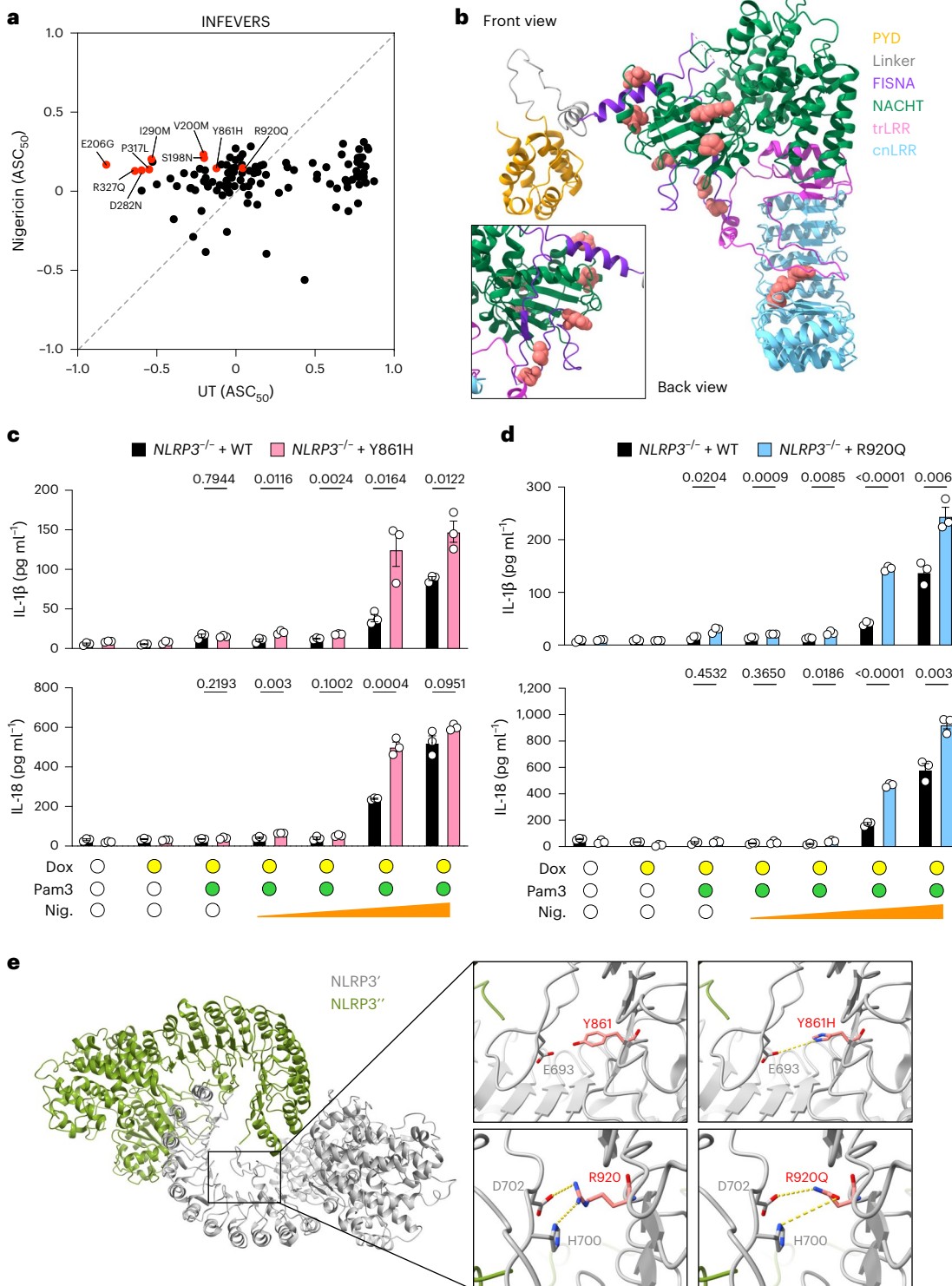

**Fig. 3 | The response of NLRP3 variants to nigericin. a**, Biplot for $ASC_{50}$ of untreated (UT) and nigericin (2 μM) treated NLRP3 variants from the INFEVERS database with $ASC_{50}$ between −1 and 1. Variants with significant increase in response to nigericin treatment are colored in red. **b**, Structural analysis of the location of nigericin-sensitive variants in the inactive NLRP3 conformation (PDB: 7PZC). **c,d**, Release of IL-1β and IL-18 from $NLRP3^{-/-}$ THP-1 cells reconstituted with WT, Y861H (**c**) or R920Q (**d**) with or without Pam3CSK4 (Pam3; 100 ng ml⁻¹) for 14 h, followed by doxycycline (Dox; 100 ng ml⁻¹) for 6 h and increasing amounts of nigericin (Nig.; 0.5 μM, 1 μM, 5 μM and 10 μM) for 1 h. **e**, Variants in the LRR domain of NLRP3 from the inactive decamer (PDB: 7PZC). Residues where variants can occur are highlighted as spheres (**b**) or sticks in red (**e**). Each dot represents a single variant (**a**) or an independent repeat (**c,d**). Data were pooled from 3–12 independent repeats (**a**, $n$ value for each variant is listed in Supplementary Table 1) or 3 independent repeats (**c,d**, mean in **a**, mean and s.e.m. in **c,d**). Two-tailed $t$-test in **c,d**.

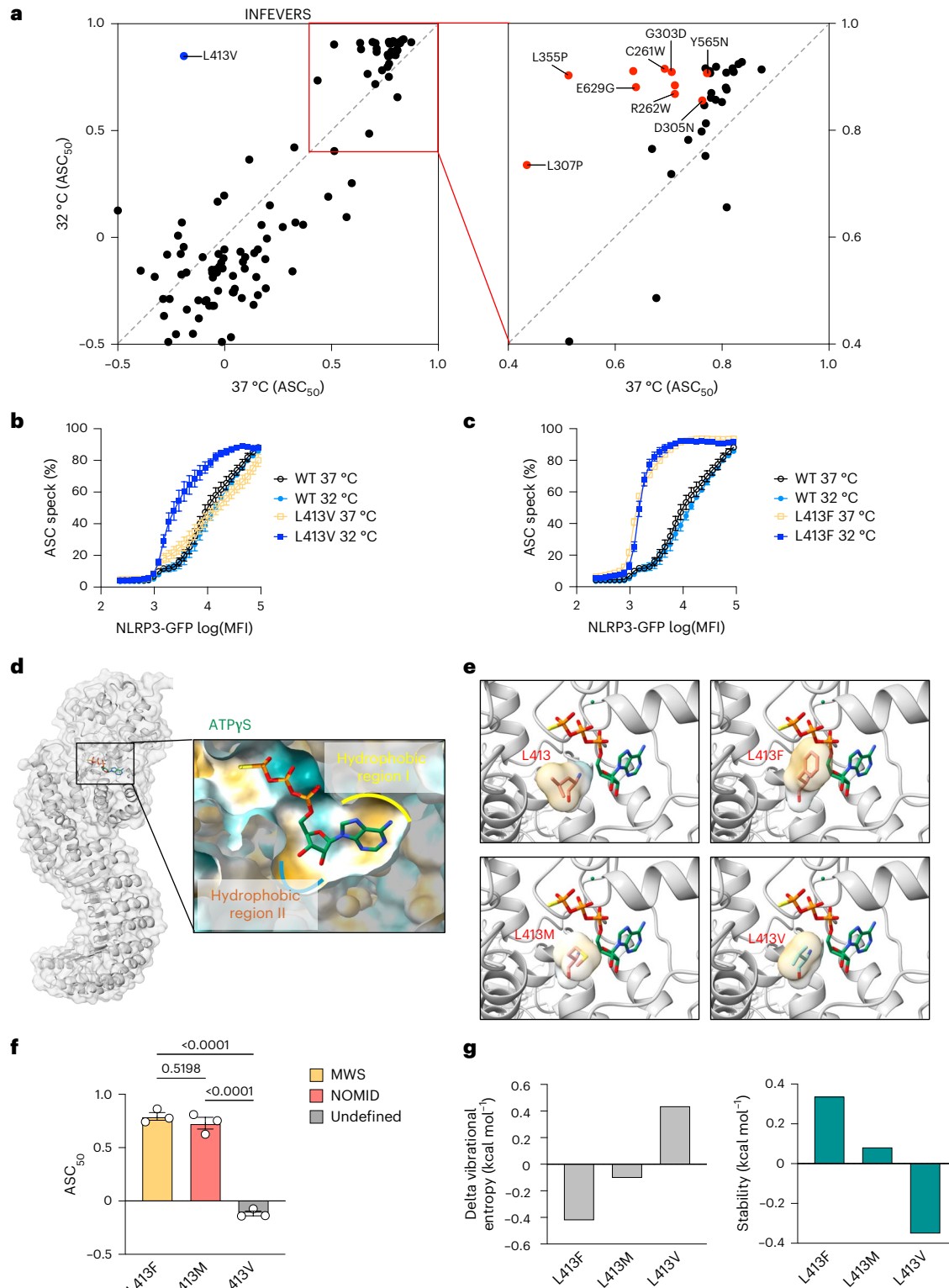

**Fig. 4 | The response of NLRP3 variants to cold temperature exposure.**
**a**, Biplot for ASC$_{50}$ of NLRP3 variants (INFEVERS database) incubated under 37 °C or 32 °C for 12 h. Left: undefined CAPS-associated NLRP3 variants, in blue. Right: FCAS-associated NLRP3 variants, in red. **b**,**c**, ASC speck percentage in the cell population with an increasing amount of NLRP3 expression for WT, L413V (**b**) or L413F (**c**) NLRP3 variants incubated under 37 °C or 32 °C. The same WT control data are used in **b** and **c**, as both were generated from the same experiment. **d**, Structural overview of the ATP-binding pocket in the active NLRP3 structure (PDB: 8EJ4). **e**, Amino acid annotation for WT, L413F, L413M and L413V in relation to ATPγS in the active NLRP3 structure (PDB: 8EJ4). The surface of the side chain

is colored based on hydrophobicity (dark cyan for most hydrophilic, and dark goldenrod for most hydrophobic). **f**, ASC$_{50}$ of untreated L413F, L413M and L413V variants colored based on associated clinical phenotype. **g**, Vibrational entropy and stability of L413F, L413M and L413V variants analyzed with DynaMut. Data were pooled from 3–12 independent repeats (**a**, *n* value for each variant is listed in Supplementary Table 1), or 3 independent repeats (**b**,**c**,**f**), each dot represents the mean value (**a**–**c**) or an independent repeat (**f**) (mean and s.e.m. in **b**,**c**,**f**). One-way ANOVA with Dunnett's multiple-comparisons test in **f**. MFI, mean fluorescence intensity.

Additionally, phosphorylation of NLRP3 at Y861 is reported to suppress the NLRP3 activity[56]. Protein tyrosine phosphatase non-receptor 22 (PTPN22) was found to dephosphorylate NLRP3 in response to inflammasome triggers, allowing efficient NLRP3 activation[56]. In our original screen, variants at this residue seemed similar to WT at baseline, although with slightly enhanced responses to nigericin stimulation. Reconstituting these variants in THP-1 cells revealed that Y861C, which cannot be inhibited by phosphorylation, has increased inflammasome activity (Extended Data Fig. 9d), whereas Y861H, which can still be phosphorylated, is similar to WT in the absence of an inflammasome trigger. Notably, the effect of Y861C was greatly diminished compared with other germline NOMID variants such as R262P. Overall, these findings are consistent with the published observation that this specific variant can drive an atypical activation signal-dependent CAPS phenotype associated with sensorineural hearing loss rather than urticarial skin manifestations[57]. This highlights the requirement for ancillary evidence when considering downgrading the classification of a specific variant based on our primary HEK293T cell screen and suggests that other complex or atypical cases may be encountered in the future requiring validation in myeloid cell lines.

### Specific NLRP3 variants are resistant to MCC950

MCC950 is a well-studied NLRP3 inhibitor[58]. Indeed, we found that the majority of the NLRP3 variants had decreased activity in response to MCC950 treatment added 4 h before analysis (Fig. 5a, Supplementary Table 1 and Supplementary Table 4). Consistent with previous findings, the NLRP3 variant L355P failed to respond to MCC950 (Fig. 5a). Additionally, we found several other hyperactive NLRP3 variants including T438P, E527V, H51R, E306K, T438I, A441P, L307P, L679P and D21H, which were insensitive to MCC950-mediated inhibition in the conditions tested. It is possible that these hyperactive NLRP3 variants could assemble into active complexes extremely rapidly after translation and thus 4 h MCC950 treatment cannot disassemble those active complexes. Therefore, we extended the MCC950 treatment time to 12 h, so it is present directly after transfection of the NLRP3 variants (Fig. 5b). Indeed, with extended MCC950 treatment time, most hyperactive NLRP3 variants had further decreased activity compared with 4 h of treatment (Fig. 5b). Notably, L355P-mediated NLRP3 hyperactivity was inhibited by MCC950 with 12 h of treatment but not 4 h (Fig. 5b); however, variants including T438P, E527V, H51R, E306K, T438I, A441P, L307P, L679P and D21H remained resistant to MCC950, suggesting a stronger tolerance of MCC950 inhibition (Fig. 5b). From the ClinVar database, we also identified that Y443C, G571E, I59V, H28Y, L29V and D60N are resistant to MCC950 (Extended Data Fig. 10a). We further compared variants which occurred at T438 and found that T438P but not T438A was resistant to MCC950-mediated inhibition at both 4-h and 12-h time points (Fig. 5c–f). Given that several of the strongly MCC950-resistant NLRP3 variants are alterations to proline residues in or near helices at the MCC950 binding site, it is possible that this introduces a higher level of structural disruption, thus impairing binding to MCC950, which normally has extensive interactions with NLRP3. Indeed, simple secondary structure predictions with JPred indicate that L307P prevents helix formation, whereas L355P did not, which may explain why this variant can be inhibited by MCC950 with longer incubation time (Extended Data Fig. 10b). Notably, the PYD domain of NLRP3 does not directly interact with MCC950 and variants in PYD are less likely to be affected by MCC950. For example, hyperactive NLRP3 variants, such as D21H, H28Y, H51R, I59V and D60N, are located in the PYD, and they remained active following MCC950 treatment (Fig. 5a,b and Extended Data Fig. 10a). We also used mCSM-lig, a bioinformatic tool[59], to analyze the changes, if any, in the affinity of NLRP3 binding to MCC950 (Fig. 5h). We found that variants including E306K, L307P, L355P, T438P, A441P and E527V had decreased predicted binding affinity to MCC950 compared with D21H and H51R, which act as negative controls (Fig. 5h). Supportively, we tested L355P, T438P/A and E527V/K

in THP-1 cells and the results are consistent with those in HEK293T cells (Extended Data Fig. 10c–e). Although most hyperactive NLRP3 variants tested in this study were sensitive to MCC950-mediated inhibition, it is important to take MCC950-insensitive variants into consideration while stratifying patients with CAPS for future clinical trials. The overall screening library of INFEVERS and ClinVar variants is available to those who would like to screen their own NLRP3 inhibitors and using our automated analysis any subsequent data generation should be highly interoperable.

## Discussion

The diagnosis of rare diseases such as CAPS is challenging. The current clinical diagnostic criterion for CAPS includes raised levels of acute-phase reactants as a mandatory criterion and the presence of two or more characteristic features such as urticarial rash, cold-triggered episodes, sensorineural hearing loss, chronic aseptic meningitis, skeletal abnormalities and musculoskeletal symptoms[60]. To corroborate the clinical diagnosis, genetic testing is also recommended for CAPS[61]; however, the same NLRP3 variant can also lead to different disease manifestations in different patients[36–40]. We analyzed all NLRP3 missense variants using a functional assay that provided strong evidence for the pathogenicity of each NLRP3 variant. We also developed the $ASC_{50}$ model that can determine the activity of NLRP3 variant activity in an unbiased manner. With the automated analysis of $ASC_{50}$ using R, our method can maximize the data representativity from smaller cell populations and can be employed for high-throughput screening assays. Most notably, our results showed that a functional test of NLRP3 variants in HEK293T cells is a time- and cost-efficient way to classify variants and thus may assist in clinical diagnosis of CAPS. Although our assay can distinguish between FCAS, MWS and NOMID in aggregate (Fig. 1c) or at a specific amino acid residue (Fig. 1d and Extended Data Fig. 3d) it did not have predictive power to determine where an individual patient would fall in the clinical spectrum of CAPS. Deep-learning models such as AlphaMissense[62] hold substantial promise to predict the pathogenicity of missense variants; however, we found it had very poor correlation to our data (Supplementary Table 1). This suggests that more experiment-based functional analysis for disease-related proteins such as NLRP3, is needed to improve future deep-learning model.

One potential caveat to our assays is that we define activation based on the expression level of NLRP3 that drives 50% ASC speck formation. Therefore, variants in NLRP3 that alter the expression level, for example by extending the half-life of NLRP3, could be missed. Fortunately, NLRP3 activation is a two-step process where priming to induce expression (signal 1) is first required before activation can take place as a second step (signal 2)[2]. Therefore, variants that alter the half-life of NLRP3 are likely to influence priming, but less likely to mimic signal 2 to trigger inflammasome formation. Additionally, priming also results in NLRP3 PTMs and relocalization that can alter NLRP3 responsiveness to an activation signal[63]. A recent study describes several NLRP3 variants which may indeed potentiate priming of the NLRP3 inflammasome and are hyperresponsive to signal 2 activators alone[30]. Notably, almost all these variants can be found in healthy individuals, and so rather than being causes of monogenic disease, they likely act as risk factors, which can potentiate disease for individuals with permissive genetic and environmental backgrounds that may bypass the threshold of activation for signal 2. Additionally, the low-penetrance variants Q705K, R490K and V200M all failed to significantly increase NLRP3 activity in our assay, in agreement with previous functional testing of Q705K and V200M[64], and the observation that at least Q705K may not be a substantial risk factor for the development of CAPS[65]. This still leaves many likely pathogenic NLRP3 variants that are clearly active in our assay, and present in small numbers of healthy individuals, which do seem to be disease-causing alleles with incomplete penetrance.

The mechanism of how NLRP3 senses K⁺ efflux remains challenging to integrate with our results. Of note, a minimal NLRP3 sequence lacking

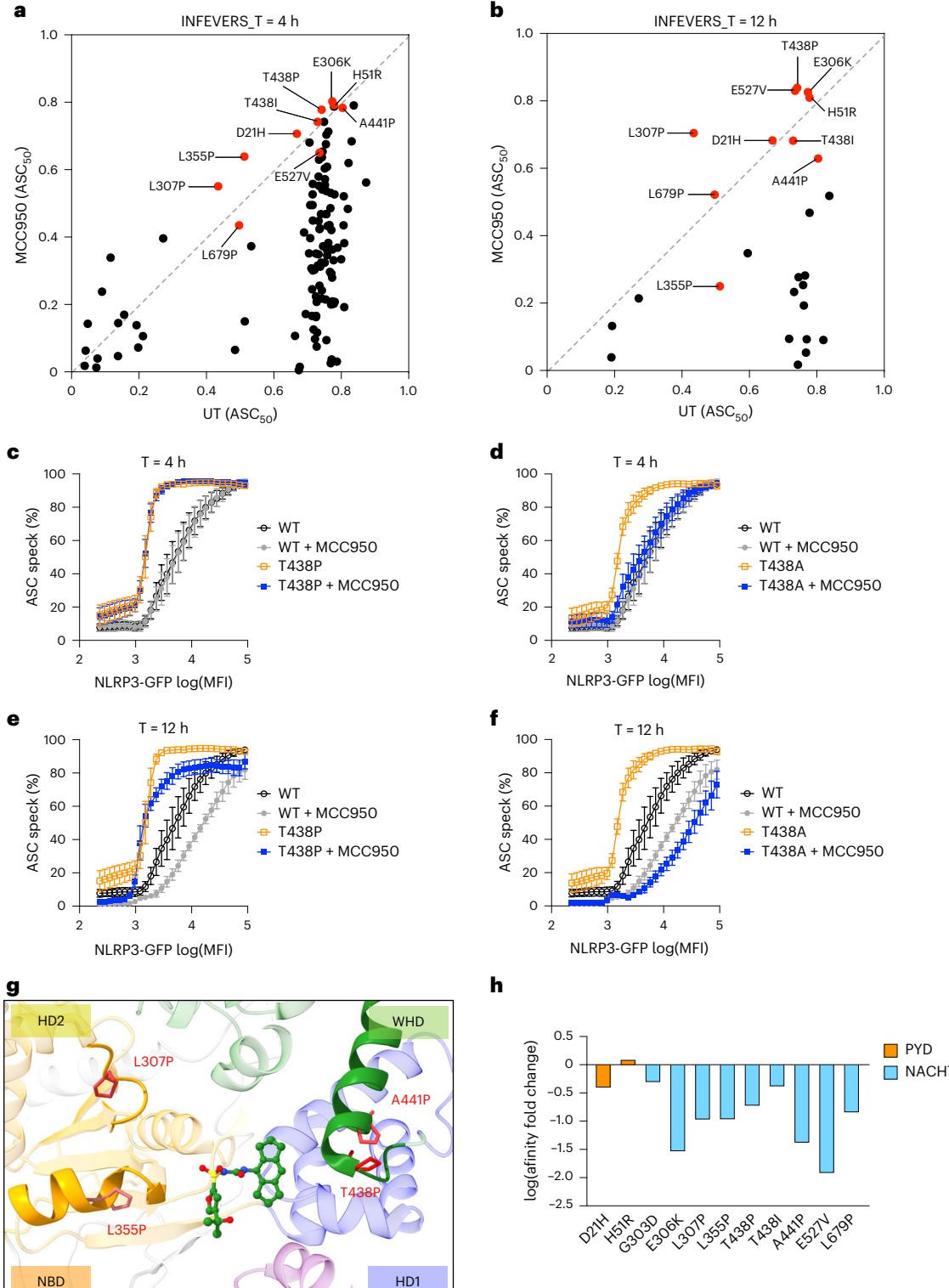

**Fig. 5 | The response of NLRP3 variants to MCC950. a,b,** Biplot for ASC₅₀ of NLRP3 variants (INFEVERS database) incubated with or without MCC950 (10 μM) for 4 h (**a**) or 12 h (**b**). **c–f,** ASC speck percentage in the cell population with an increasing amount of NLRP3 expression for WT or T428P or T438A NLRP3 variant with or without MCC950 (10 μM) treatment for 4 h (**c,d**) or 12 h (**e,f**). **g,** Structural annotation of variants that can disrupt the MCC950-binding pocket in the inactive NLRP3 structure (PDB: 7PZC). **h,** Prediction of ligand affinity between inactive NLRP3 (PDB: 7PZC) and MCC950 (PDB: 8GI) using mCSM-lig. Variants located in the PYD are colored in orange. Variants located in the NACHT domain are colored in blue. The same control data are used when results were generated from the same experiment (**c–f**). Data were pooled from 3–12 independent repeats (**a**, *n* value for each variant is listed in Supplementary Table 1) or 3 independent repeats (**c–f**), each dot represents the mean value of ASC₅₀ for each NLRP3 variant (**a,b**) or mean and s.e.m. (**c–f**).

the LRR can still be activated by K⁺ efflux similar to the full-length NLRP3 (ref. 66). Inactive, ADP-bound NLRP3 forms a decamer via intertwined LRR domains[5]. How the LRR-deficient NLRP3 maintains its inactive status is unclear. On the contrary, we found that the LRR domain harbors variants that sensitize NLRP3 for nigericin-induced K⁺ efflux. Three variants, R920Q, Y861H and Y861C occurred at the direct interface between the acidic loop and the concave side of the LRR. Supportively, a previous study found that monocytes from patients with CAPS carrying a variant at Y861, required an activation signal to produce elevated IL-1β compared with those from healthy donors[57]. In the inactive NLRP3 decameric cage, the acidic loop folds onto the basic patch in the concave side of the LRR, mediating contacts with the adjacent NLRP3 subunit[5]. Variants affecting the positioning of this acidic loop might destabilize the NLRP3 inactive complex and promote NLRP3 activity; however, Y861H and R920Q did not increase the NLRP3 activity to a high level as those in the NACHT domain. It is possible that a trigger, such as nigericin, is still required to induce conformational changes in these NLRP3 variants and organize the protein translocation among subcellular structures. Moreover, we show that variants in the acidic loop also led to NLRP3 hyperactivation. Overall, our data agree best with a mechanism of K⁺ efflux-induced NLRP3 activation that might involve repositioning of the acidic loop, which then releases NLRP3 from its inactive decameric cage.

Low temperature as a trigger for symptoms in patients with FCAS is also an area of ongoing mechanistic research. There has been some speculation that this promotes the stability of the protein, increasing its expression levels in the cell. As described above, we believe that this is unlikely as it would mostly be restricted to an effect on signal 1, which is dispensable from our assay. Furthermore, we have clearly identified temperature-sensitive variants associated with FCAS in our assay, whose ASC$_{50}$ occurs at far lower NLRP3 levels compared with WT. A previous study showed that cryosensitive NLRP3 variants, including L355P and R262W, had reduced binding affinity with the temperature-sensitive protein, heat shock cognate 70 (HSC70) when exposed to low temperature[67]. Knockdown of HSC70 leads to increased activity of L355P and R262W NLRP3 variants[67]; however, L413V, L355P and R262W are in the core of the NACHT domain with low surface accessibility. Instead, we have found one extraordinarily temperature-sensitive variant, L413V, which seems to influence the stability and flexibility of the ATP-binding site. Therefore, we suggest that one way the other temperature-sensitive variants in NLRP3 may operate is via this same mechanism at the ATP-binding site. Although it is not immediately obvious at the structural level how these variants could have such an effect at a distance, that is also consistent with their reduced temperature sensitivity compared with L413V. Further studies could help address separate mechanisms underlying temperature sensitivity of NLRP3 variants or if there is a unifying principle for them all.

Pharmacological inhibition of NLRP3 activation has been shown to reduce inflammation in many inflammatory disease models in rodents[13,68]. Although anti-IL-1 therapies, including anakinra, canakinumab and rilonacept are safe and efficient therapies to treat CAPS[14,15], small molecules targeting NLRP3 have many advantages over biologicals. Our study investigated the inhibitory efficacy of one such molecule, MCC950, on hyperactive NLRP3 variants from INFEVERS or ClinVar databases. While most of the hyperactive NLRP3 variants can be efficiently inhibited by MCC950, a small group of variants were found to be resistant, even after prolonged incubation. Given that there are few such variants in NLRP3, future clinical trials could avoid including patients who may have less-durable responses to MCC950, and molecules with different binding sites could eventually be developed. One caveat is that our screening assay expresses only the variant NLRP3 protein, whereas patients with CAPS also express one WT allele of NLRP3. Therefore, it is possible that an MCC950-resistant NLRP3 variant could still benefit from inhibition of the WT NLRP3 allele if they both participate in the formation of an active inflammasome complex.

In summary, our research provided functional results to assist diagnosis of patients with CAPS, a deeper understanding of the activation mechanisms of NLRP3 and a basis for categorizing patients for the potential use of targeted treatments in the future.

## Online content

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

[1]Centre for Innate Immunity and Infectious Diseases, Hudson Institute of Medical Research, Clayton, Victoria, Australia. [2]Department of Molecular and Translational Science, Monash University, Clayton, Victoria, Australia. [3]Inflammation Division, The Walter and Eliza Hall Institute of Medical Research, Parkville, Victoria, Australia. [4]Department of Medical Biology, University of Melbourne, Parkville, Victoria, Australia. [5]Immunology Division, The Walter and Eliza Hall Institute of Medical Research, Parkville, Victoria, Australia. [6]Institute of Structural Biology, Medical Faculty, University of Bonn, Bonn, Germany. [7]Department of Biochemistry and Molecular Biology, Immunity Program, Biomedicine Discovery Institute, Monash University, Clayton, Victoria, Australia. [8]Department of Molecular genetics and Cytogenomics, CHU Montpellier, Rare and Autoinflammatory Diseases Unit, University of Montpellier, CEREMAIA, Institute for Regenerative Medicine and Biotherapy, INSERMU1183, Montpellier, France. [9]Queensland Paediatric Immunology and Allergy Service, Children's Health Queensland, Brisbane, Queensland, Australia. [10]Department of Immunology, Hospital Clínic, Barcelona, Spain. [11]Institut d'Investigacions Biomèdiques August Pi i Sunyer, Barcelona, Spain. [12]School of Medicine, University of Barcelona, Barcelona, Spain. [13]Sectional Head Pediatric Rheumatology, SRCC Children's Hospital, Mumbai, India. ✉e-mail: seth.masters@hudson.org.au

# Methods

## Plasmids and site-directed mutagenesis

The mammalian expression plasmid, peGFP-C2 encoding human NLRP3 (NM_004895, Addgene, plasmid #73955) tagged with N-terminal eGFP was a gift from C. Stehlik[69]. The pcDNA3-N-HA-NEK7 K64M was a gift from B. Beutler (Addgene, plasmid #75143) and was modified in this study[44]. The mCherry sequence was cloned into pcDNA3-N-HA-NEK7 K64M between the HindIII and BamHI to generate mCherry-tagged NEK7$^{K64M}$. Guide RNAs (gRNAs) targeting NLRP3 or NEK7 were cloned into lentiCRISPR v2 (Addgene, plasmid #52961)[70] (Supplementary Table 6). GFP-NLRP3 was cloned into the doxycycline-inducible lentiviral construct, pF-TRE3G_hygro (gifted from J. Silke) between BamHI and NheI. Whole plasmid sequencing was performed to confirm correct insertion and the absence of unwanted mutations. Point mutations were introduced by QuikChange mutagenesis (catalogue number 210514, Agilent Technologies) according to the manufacturer's instructions. Primers used are listed in Supplementary Table 6. The NLRP3 variants library was produced by Genscript Biotech. All NLRP3 variant sequences were confirmed using Sanger sequencing.

## Cell culture

Human embryonic kidney (HEK) 293T cells (ATCC CRL-3216) stably expressing BFP-tagged ASC were generated through retroviral transduction[71]. WT and BFP-ASC-expressing HEK293T cells were cultured in a humidified incubator at 37 °C and 10% $CO_2$ in Dulbecco's modified Eagle's medium (Thermo Fisher Scientific) supplemented with 10% fetal calf serum, 2 mM L-glutamine (Gibco, 25030081), 100 µg ml$^{-1}$ streptomycin, and 100 U ml$^{-1}$ penicillin (Gibco, 1514022). Human myeloid cell line, THP-1 cells (ATCC TIB-202), U937 cells were cultured in a humidified incubator at 37 °C and 5% $CO_2$ in RPMI 1640 medium (Thermo Fisher Scientific) supplemented with 10% fetal calf serum, 2 mM L-glutamine (Gibco, 25030081) and 100 µg ml$^{-1}$ streptomycin, and 100 U ml$^{-1}$ penicillin (Gibco, 1514022). Human PBMCs were sourced from the Victorian Blood Donor Registry under WEHI HREC approval 18/07.

## Generation of stable cell lines

To generate HEK293T cells stably expressing ASC-BFP, 2 µg retroviral plasmid, 1 µg gag-pol plasmid and 100 ng VSVg plasmid were transfected into $1.5 × 10^6$ WT HEK293T cells seeded in a 10-cm dish using LipofectMax following manufacturer's instructions (ABP Biosciences, FP311) to make retroviral particles. To generate knockout cell lines, gRNAs were designed using the online tool CHOPCHOP (http://chopchop.cbu.uib.no/) and cloned into the lentiCRISPR v2 plasmid (Addgene, #52961)[72]. For generation of lentiviral particles, 10 µg gRNA construct, 5 µg lentiviral packaging vector, pMDL, 2.5 µg RSV-REV and 3 µg VSVg were transfected into WT HEK293T cells as described above. Transfected cells were changed into fresh medium 12 h post-transfection. After 36 h, the viral supernatants were collected and filtered using 0.45-µm filters. One million WT THP-1 cells or WT HEK293T cells were resuspended in 3 ml viral supernatant and supplemented with 8 µg ml$^{-1}$ Polybrene. The mixture was seeded in a six-well plate and centrifuged at 1,017$g$ for 3 h before returning to the incubator. Cells were replaced with fresh medium the next day and incubated for 3 days until selection with 1 µg ml$^{-1}$ puromycin (InvivoGen, ant-pr-1) for 3 days. Immunoblotting and inflammasome assays were performed to confirm the successful knockout of NLRP3 or NEK7 or both.

To express NLRP3 variants in THP-1 cells, lentivirus was packaged using 7.5 µg pF-TRE3G_hygro plasmid containing gene of interest, 5 µg pCMV delta R8.2 and 5 µg VSVg into WT HEK293T cells as described above. Cells were selected using 0.5 mg ml$^{-1}$ hygromycin (Gibco, 10687010).

## Immunoblotting

One million cells for each genotype were resuspended in 100 µl RIPA buffer (20 mM Tria-HCl, 150 mM NaCl, 1 mM EDTA, 1% Triton X-100, 0.5% sodium deoxycholate, 3.5 mM sodium dodecyl sulfate and 10% glycerol) supplemented with protease inhibitor (Roche, 11697498001) and 1 mM phenylmethylsulfonyl fluoride (Roche, 10837091001). Cell lysates were centrifuged at 14,000$g$ for 15 min at 4 °C, and supernatant was collected and mixed with 4× SDS sample buffer before heated at 96 °C for 15 min. Proteins were separated on 4–12% Bolt Bis-Tris Plus gels (Thermo Scientific, NW0412F) and transferred to polyvinyl difluoride membranes (Millipore, IPVH00010). Membranes were blocked in 5% skim milk in Tris-buffered saline with 0.1% Tween-20 (TBST) and incubated overnight with antibodies against NLRP3 (1:1,000 dilution, rabbit, CST, 15101S), NEK7 (1:1,000 dilution, rabbit, CST, 3057S), tubulin (1:5,000 dilution, rat, Santa Cruz Biotechnology, sc-53029) or β-actin conjugated with horseradish peroxidase (1:5,000 dilution, Santa Cruz Biotechnology, SANTSC-47778HRP). Polyvinyl difluoride membranes were then incubated with anti-rabbit (1:5,000 dilution, Invitrogen, 31460) or anti-rat (1:5,000 dilution, Invitrogen, 31470) horseradish peroxidase-conjugated secondary antibodies for 1 h and proteins were visualized using Clarity Western ECL Substrate (170-5061, Bio-Rad) and the ChemiDoc Touch Imaging System (Bio-Rad). Immunoblots were analyzed using ImageLab Software v.6.01.

## ELISA

Cytokine concentrations from THP-1 cells were measured using IL-1β (R&D, DY201), IL-18 (R&D, DY318-05) or IL-6 (R&D, DY206) ELISA according to the manufacturers' instructions. Data were collected using a FLUOstar OPTIMA (BMG Labtech) plate reader and analyzed using MARS (v.3.01 R2).

## Imaging flow cytometry

HEK293T cells stably expressing BFP-ASC were transfected with plasmids expressing GFP-NLRP3. Cells were collected and fixed in prechilled methanol and resuspended in PBS supplemented with 2% FCS and 5 mM EDTA. Samples were acquired using an Amnis ImageStreamX MKII imaging flow cytometer[34]. Data were analyzed using IDEAS software (v.6.2).

## RT−PCR

RNA was isolated using the Isolate II RNA Mini kit (Meridian Bioscience, BIO-52073) as per the manufacturer's instructions. The isolated RNA was converted into complementary DNA using SuperScript III Reverse Transcriptase (Invitrogen, 18080093) following the manufacturer's instructions. The qPCR primers are listed in Supplementary Table 6. PCR with reverse transcription (RT−PCR) was performed on the QuantStudio 12K Flex instrument with Maxima SYBR Green Real Time PCR Master Mixes (Thermo Fisher, K0253) analyzed by Quanstudio Design and Analysis Software (v.2.8).

## Transfection and stimulation of cells

ASC-BFP HEK293T cells were seeded at $2 × 10^4$ cells per well in 96-well plates and allowed to rest overnight. Cells were transfected with 50 ng plasmid DNA using Lipofectamine 2000 (catalogue number 11668–019, Life Technologies) in OptiMEM I (catalogue number 11058021, Gibco) following the manufacturer's instructions. To induce K$^+$ efflux, transfected cells were incubated for 11 h and then treated with 2 µM of nigericin (catalogue number tlrl-nig, InvivoGen) for 1 h. To inhibit K$^+$ efflux, KCl was added 30 min before nigericin treatment. Cells were then collected using trypsin-EDTA (catalogue number 10006132) and fixed in pre-chilled methanol before flow cytometry analysis. For the cold temperature exposure experiment, transfected cells were incubated at either 32 °C or 37 °C in a 10% $CO_2$ incubator for 12 h. Cells were collected using trypsin-EDTA and fixed in pre-warmed 4% PFA (catalogue number 15710, ProSciTech). To generate ASC-BFP HEK293T cells expressing mCherry-tagged NEK7$^{K64M}$, cells were transfected with pcDNA3-N-mCherry-NEK7 K64M for 14 h and then sorted for mCherry-positive cells using a flow cytometer. These cells were then

seeded into a 96-well plate and were transfected with plasmids encoding different NLRP3 variants the following day for 12 h as described above. For the inhibition assay, transfected cells were treated with 10 μM MCC950 (catalogue number inh-mcc, InvivoGen) for 4 or 12 h and cells were collected 12 h post-transfection. After fixation, all samples were kept at 4 °C until analysis.

THP-1 cells were seeded at $1 \times 10^5$ cells per well in 96-well plates. To activate the NLRP3 inflammasome, cells were primed with 100 ng ml$^{-1}$ Pam3CSK4 (InvivoGen, tlrl-pms) for 14 h and then treated with nigericin (InvivoGen, tlrl-nig), amount and duration as indicated in figure legends. To activate the pyrin inflammasome, cells were primed with primed with Pam3CSK4 (100 ng ml$^{-1}$) and IFN-γ (1 μg ml$^{-1}$, R&D systems, 285-IF-100) for 24 h and then treated with TcdB (200 ng ml$^{-1}$, Abcam, ab124001) for 14 h. To induce THP-1 differentiation, cells were treated with 100 ng ml$^{-1}$ phorbol-12-myristat-13-acetat (PMA, Sigma-Aldrich, 524400) for 14 h and rested in fresh cell culture medium for 24 h before priming with 50 ng ml$^{-1}$ LPS (Enzo, ALX-581-010) for 3 h. After priming GFP-NLRP3 expression was induced by treating cells with doxycycline for 6 h before stimulation with inflammasome activators. To induce inflammasome activation, THP-1 monocytes or PMA-induced macrophages were treated with nigericin (amount as in the figure legend) (InvivoGen, tlrl-nig) for 1 h or 200 μg ml$^{-1}$ MSU (InvivoGen, tlrl-msu), 50 μM imiquimod (ChemSupply, 99011-02-6), 50 μg ml$^{-1}$ LLOMe (Sigma-Aldrich, L7393) for 14 h. To inhibit NLRP3 inflammasome activation in THP-1 cells, cells were treated with 10 μM MCC950 together with doxycycline treatment after priming. After treatment, cell culture supernatants were collected for ELISA and cells were resuspended in PBS supplemented with 2% FCS and 5 mM EDTA to measure the percentage of GFP-positive cells using a Cytek Aurora flow cytometer.

### Flow cytometry and automated analysis pipeline

Cells were acquired with BD LSR Fortessa X20 or Cytek Aurora cytometers. Data were analyzed using BD FACSDiva Software (v.9.0) SpectroFlo (v.3.2.1) or FlowJo (v.10.8.2). FCS files were first imported into the RStudio (v.2024.04.2+764) environment using the package flowCore[73], followed by a data quality control step using the package flowAI[74], which filters outlier data points based on discrepancies in flow rate, signal acquisition and dynamic range. An automated gating pipeline was then implemented using openCyto[75], to first gate for non-debris single cells and then gate on ASC speck-positive and -negative populations (Extended Data Fig. 1f). To ensure consistency, gates were set based on a WT control sample, then applied across the sample set.

Following gating, single-cell fluorescence values of GFP-NLRP3 expression were exported for each population. The proportion of speck-positive cells of the total population was calculated using a step-binning method, in which a bin of fixed width (one-quarter the range of the MFI) was 'stepped' through the data, moving forward toward increasing GFP-NLRP3 expression in 0.1 ΔMFI increments. Within each bin step, the number of speck-positive and -negative cells was compared, with the proportion of positive cells calculated. This process allows for the generation of a dose–response curve (Extended Data Fig. 2f) based on the level of NLRP3 expression in a cell.

Relative $EC_{50}$ values for each curve were generated using the package 'drda'[76]. We utilized a four-parameter logistic function 'L4' with the following limits (min,max): $\alpha$ (0,1), $\delta$ (−inf,1), $\eta$ (−inf,+inf) and $\varphi$ (0,+inf), where $\alpha$ is the value of the function when $x$ approaches infinity, $\delta$ is the height of the curve, $\eta$ is the steepness of the curve and $\varphi$ is the midpoint ($EC_{50}$). Obtained $EC_{50}$ values for each sample were converted into $ASC_{50}$ using the following equation: $(EC_{50}(\text{WT control}) - EC_{50}(\text{sample}))/EC_{50}(\text{WT control})$.

For analysis using FlowJo, cells were then gated on the BFP-ASC and GFP-NLRP3 double-positive population. This population was further divided into eight subgroups based on GFP-NLRP3 expression. The ASC speck percentage was analyzed for each subgroup individually.

### Bioinformatic analysis

DynaMut and DDmut were used to predict protein structural stability after mutation[50,77]. The mCSM-lig was used to predict ligand affinity between NLRP3 and MCC950 (ref. 59). Jpred was used to predict the secondary structure[78].

### Structural analysis

The structure of the active NLRP3 inflammasome disk (PDB: 8EJ4), the inactive MCC950-bound NLRP3 decamer (PDB:7PZC), the NEK7-bound active NLRP3 (PDB: 6NPY) and the PYD filament of NLRP3 (PDB: 7PZD) have been described previously[5,11,12,42]. Structural analysis and point mutations were performed using ChimeraX. The orientation of a single amino acid variant with the least number of clashes was chosen. The $ASC_{50}$ data were annotated into the NLRP3 structure with Pymol (v.2.5.0, Schrödinger) using the data2bfactor script written by R.L. Campbell. The Python script for data2bfactor was updated by ChatGPT (OpenAI) and adapted to the latest version of Python.

### Statistical analysis

Statistical significance was calculated using GraphPad Prism v.9.5.1. Statistical tests used are stated in the respective figure legends. $P$ values are labeled in each figure. For each table, a two-tailed $t$-test was used to compare untreated and treated variant. Number of independent repeats ($n$) for each variant are listed in Supplementary Tables 1–4.

### Reporting summary

Further information on research design is available in the Nature Portfolio Reporting Summary linked to this article.

## Data availability

Our data are deposited in publicly available data repositories, including INFEVERS (identifier NLRP3, https://infevers.umai-montpellier.fr) and GenIA (identifier NLRP3, https://www.geniadb.net/app/ref/info.php?id=1380). Databases used in this study include INFEVERS (identifier NLRP3, https://infevers.umai-montpellier.fr/web/), ClinVar (identifier NLRP3, https://www.ncbi.nlm.nih.gov/clinvar/), AlphaMissense (identifier NLRP3, https://alphamissense.hegelab.org/search) and gnomAD (v.4.1.0, identifier NLRP3, https://gnomad.broadinstitute.org/gene/ENSG00000162711?dataset=gnomad_r4). Datasets used in this study are available in the RCSB Protein Data Bank (https://www.rcsb.org) with accession codes PDB 8EJ4, 7PZC, 6NPY and 7PZD. Source data are provided with this paper.

## Code availability

The code used to perform the data analysis in this paper is available in a GitHub repository (https://github.com/Seth-Masters-Lab/Speck-Assay-Automated-Workflow.git) and is publicly available. Additional codes used are also publicly available including flowCore (release number bioc_3.13, https://github.com/RGLab/flowCore), flowAI (release number flowise@1.4.0, https://github.com/FlowiseAI/Flowise), openCyto (release number bioc_3.13, https://github.com/RGLab/openCyto), drda (Malyutina et al. 2023, https://github.com/albertopessia/drda) and data2bfactor (2003 Campbell, https://peterslab.org/downloads.php).

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

## Acknowledgements

We acknowledge the facilities and the technical assistance of the flow cytometry laboratory at Walter and Eliza Hall Institute of Medical Research, FlowCore and Monash Micro Imaging at the Monash Health Translation Precinct and FlowCore at the Alfred Research Alliance. S.L.M. was supported by the European Joint Programme on Rare Diseases (ODINO project, EJP RD JTC2022) and the NHMRC (GNT2008699). S.F. was supported by the Jack Brockhoff Foundation (grant number 5252). D.D.N. was supported in part by a Monash University FMNHS Senior Postdoctoral Fellowship. M.G. is supported by the European Research Council (Advanced Grant NalpACT) and by the German Research Foundation (DFG) under Germany's Excellence Strategy—EXC2151-390873048. M.Z. is supported by the DFG-funded International Graduate School GRK 2168.

## Author contributions

S.F. and S.L.M. conceived the study. S.F., T.R., A.S., Y.Z., A.D., D.D.N., M.Z. and D.L.N. designed or performed experiments. S. F., M.W. and K.H.-S. analyzed the data. P.J.B., M.G., F.M. and G.B. provided technical support. A.P., J.I.A. and R.P.K. conducted clinical investigations. S.F., M.W. and S.L.M. wrote the paper. All authors commented on the paper. S.L.M. supervised the study.

## Funding

## Competing interests

S.L.M. is the Vice President of Discovery Biology of NRG Therapeutics and scientific advisor for Odyssey Therapeutics. M.G. is a scientific advisor to BioAge Labs. The other authors declare no competing interests.

## Additional information

**Extended data** is available for this paper at https://doi.org/10.1038/s41590-025-02088-9.

**Correspondence and requests for materials** should be addressed to Seth L. Masters.

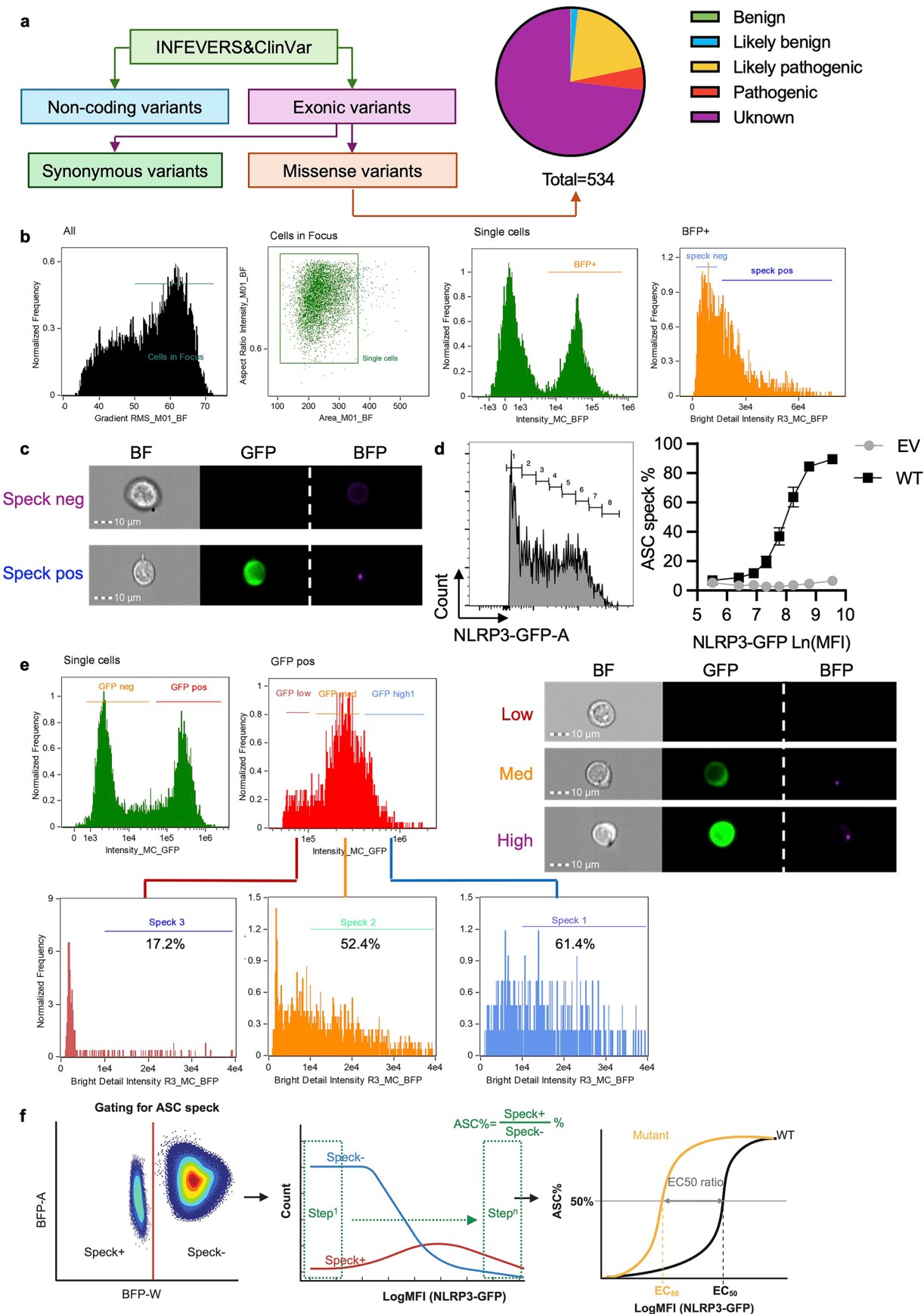

**Extended Data Fig. 1 | See next page for caption.**

**Extended Data Fig. 1 | ASC speck percentage is increased in cell populations with high NLRP3 expression. a**. Schematic diagram of NLRP3 variants and classification. **b**. Gating strategy for image-based flow cytometry analysis of ASC speck formation in ASC-BFP HEK293T cells. **c**. Representative images for ASC speck-negative and positive cells following gating as in b. **d**. Flow cytometry analysis of ASC speck percentage in cell groups with different levels of NLRP3 expression. **e**. Gating strategy (left) and representative images (right) for ASC speck percentage in cell populations with low, medium, or high level of NLRP3-GFP expression. Images were created using BioRender. **f**. Schematic diagram of automated ASC50 analysis progress. Cells were separated into a speck-positive and speck-negative population (left) and then analyzed based on the NLRP3-GFP signal (middle). The ASC percentage was calculated for each population with the same level of GFP expression. The EC50 was calculated based on the ASC percentage curve and the EC50 ratio was generated by comparing the mutant to wild-type (WT) NLRP3 (right). Data were pooled from 4 independent repeats (**d**, mean and s.e.m. in **d**) or representative of 3 independent repeats (**b**, **c**, **e**).

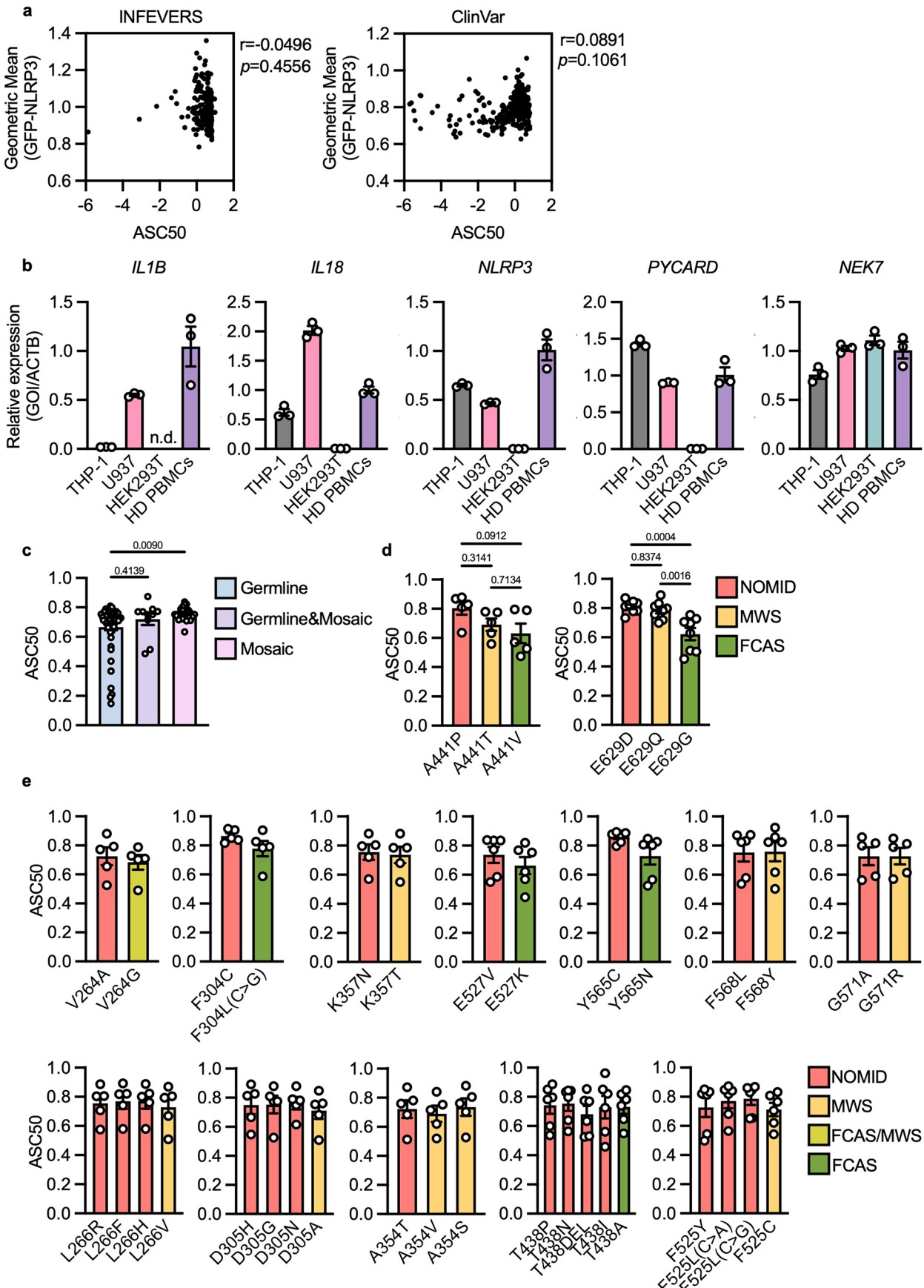

**Extended Data Fig. 2 | See next page for caption.**

**Extended Data Fig. 2 | Analysis of NLRP3 variants using ASC50. a**. Correlation analysis for the association between the geometric mean which represents GFP-NLRP3 level and the ASC50 value for variants from INFEVERS and ClinVar database. **b**. Expression of human *IL1B*, *IL18*, *NLRP3*, *PYCARD* and *NEK7* in THP-1, U937, HEK293T and healthy donor peripheral blood mononuclear cells (PBMCs) relative to *ACTB*. **c**. ASC50 of pathogenic and likely pathogenic variants that are germline, germline/mosaic or mosaic. **d**. ASC50 of NLRP3 with variants at A441 or E629 associated with different disease presentations. **e**. ASC50 of NLRP3 with variants identified at the same site associated with different disease presentations. NOMID, neonatal-onset multisystem inflammatory disease; MWS, Muckle–Wells syndrome; FCAS, familial cold autoinflammatory syndrome. Data are representative of 3 independent repeat (**a**) or collective of 3 independent repeats (**b**) or 5–11 independent repeats (**c**-**e**, *n* value for each variant is listed in Supplementary Table 1). One-way ANOVA with Dunnett's multiple-comparisons test in **c**, **d**.

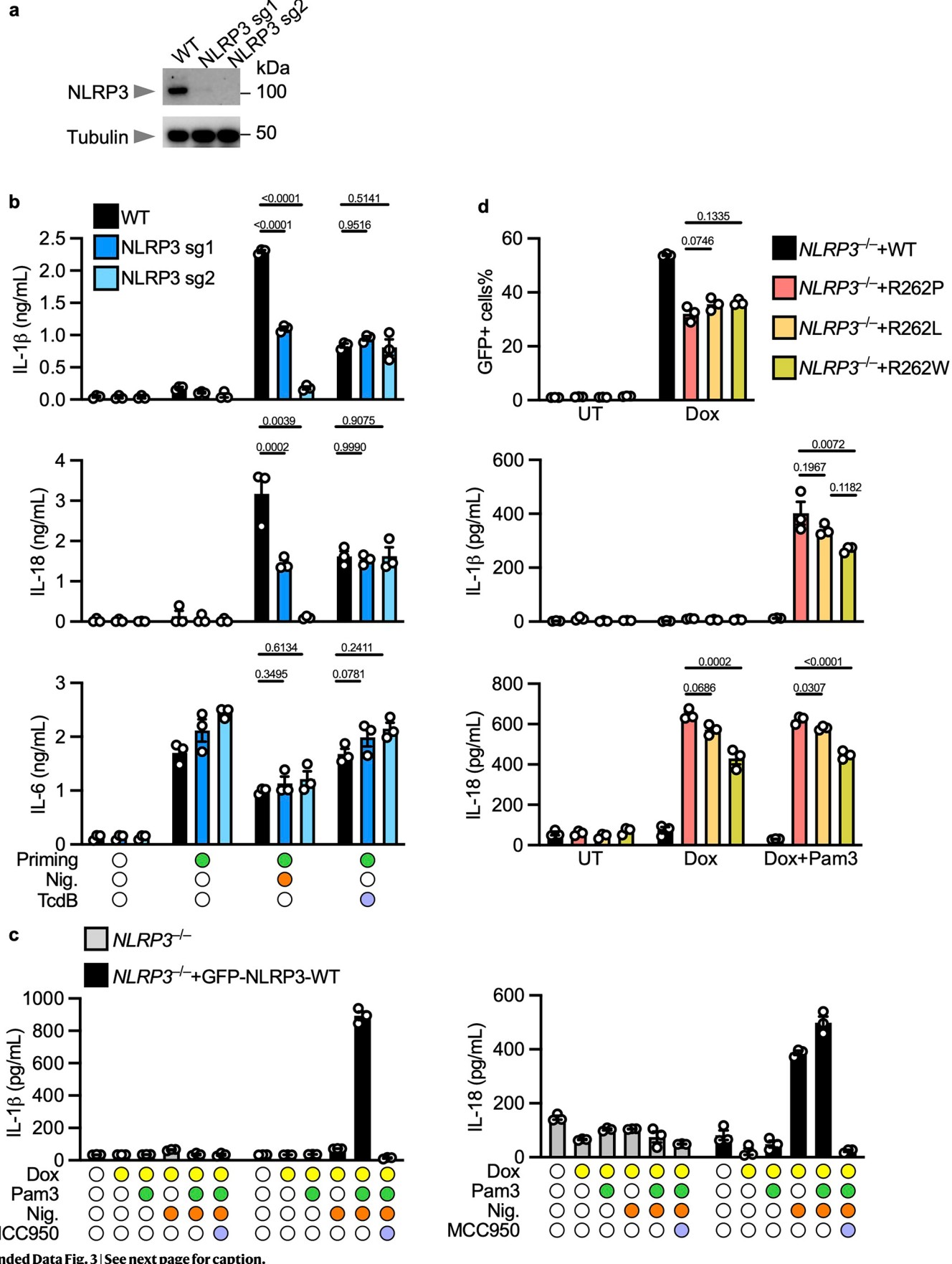

**Extended Data Fig. 3 | See next page for caption.**

**Extended Data Fig. 3 | NLRP3 reconstitution in THP-1 cells. a.** Immunoblot analysis of NLRP3 and tubulin in WT THP-1 cells or NLRP3 CRISPR-knockout using guide 1 or 2 (sg1 or 2). **b.** Release of IL-1β, IL-18 and IL-6 from *NLRP3*$^{-/-}$ THP-1 cells left untreated, or primed with Pam3CSK4 (Pam3, 100 ng/mL) for 14 hours and then with nigericin (Nig., 10 μM) for 3 hours or primed with Pam3CSK4 (Pam3, 100 ng/mL) and IFN-γ (1 μg/mL) for 24 hours and then treated with TcdB (200 ng/mL) for 14 hours. **c.** Release of IL-1β and IL-18 in *NLRP3*$^{-/-}$ or *NLRP3*$^{-/-}$ THP-1 cells reconstituted with GFP-tagged wild-type (WT) NLRP3 left untreated or treated with Pam3CSK4 (Pam3, 100 ng/mL) for 14 hours followed by doxycycline (Dox, 100 ng/mL) for 6 hours with or without MCC950 (10 μM), and then nigericin (Nig., 10 μM) for 1 hour. **d.** Percentage of cells expressing GFP and the release of IL-1β and IL-18 in *NLRP3*$^{-/-}$ THP-1 cells reconstituted with GFP-tagged wild-type (WT), R262P, R262L or R262W NLRP3 after treatment with Pam3CSK4 (Pam3, 100 ng/mL) for 14 hours and doxycycline (Dox, 100 ng/mL) for 6 hours. Data are representative of two independent repeats (**a**) or collective of three independent repeats (**b-d**, mean and s.e.m. in **b-d**). One-way ANOVA with Dunnett's multiple-comparisons test in **b**, **d**.

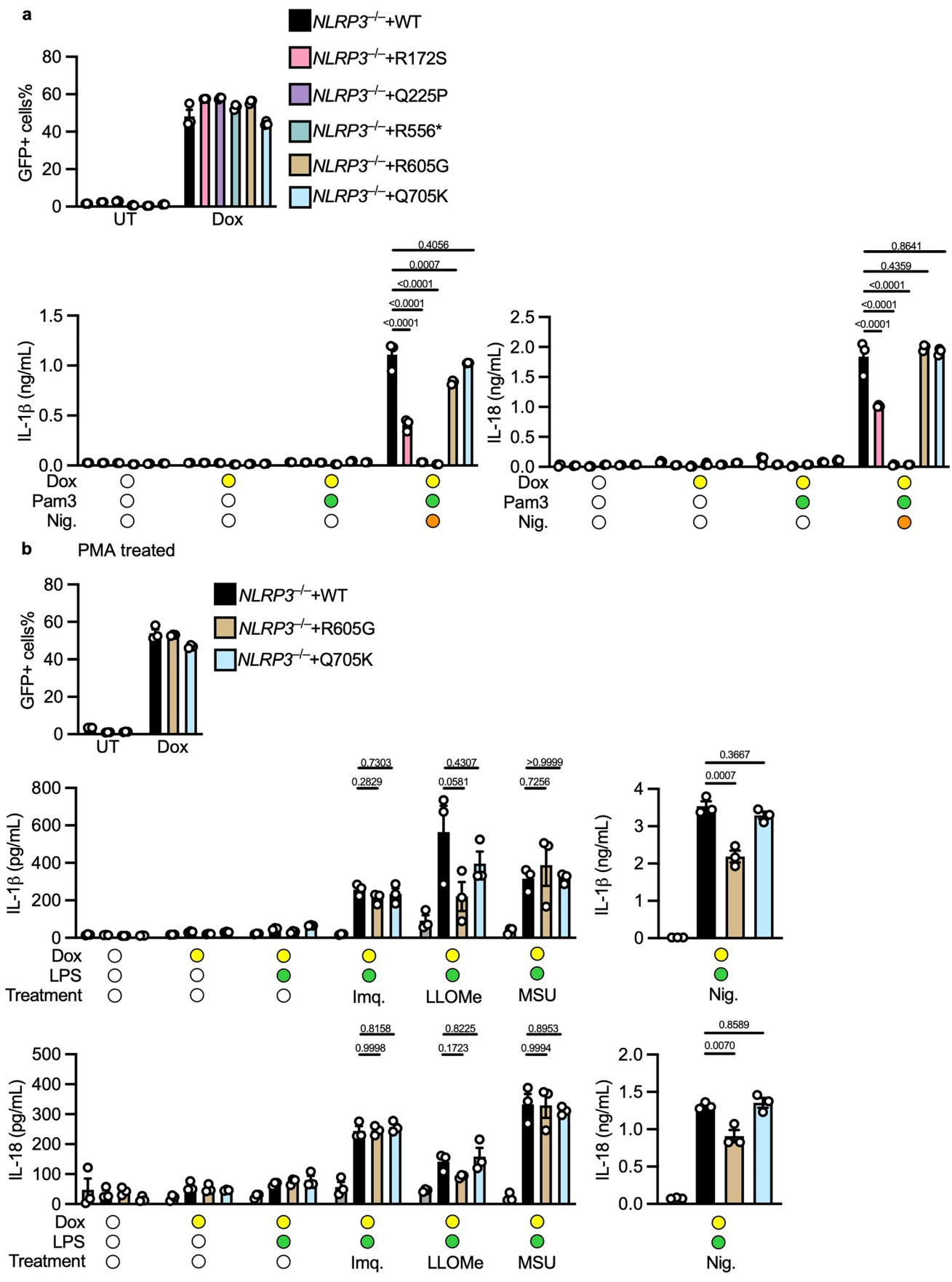

**Extended Data Fig. 4 | See next page for caption.**

**Extended Data Fig. 4 | Inflammasome activity of NLRP3 variants of uncertain significance in reconstituted THP-1 cells. a**. Percentage of cells expressing GFP and the release of IL-1β and IL-18 in *NLRP3*[−/−] THP-1 cells reconstituted with GFP-tagged wild-type (WT), R172S, Q225P, R556*, R605G or Q705K NLRP3. Cells were either untreated or primed with Pam3CSK4 (Pam3, 100 ng/mL) for 14 hours followed by doxycycline (Dox, 100 ng/mL) for 6 hours, and then nigericin (Nig., 10 μM) for 1 hour. **b**. Percentage of cells expressing GFP and the release of IL-1β and IL-18 in PMA-induced, differentiated *NLRP3*[−/−] THP-1 cells reconstituted with GFP-tagged WT, R605G or Q705K NLRP3. Cells were either untreated or primed with LPS (50 ng/mL) for 3 hours followed by treatments with NLRP3 activators including imiquimod (Imq., 50 μM), LLOMe (50 μg/mL), monosodium urate crystals (MSU, 200 μg/mL) for 14 hours and nigericin (Nig., 10 μM) for 1 hour. Data were pooled from three independent repeats (**a-b**). One-way ANOVA with Dunnett's multiple-comparisons test in **a**, **b**.

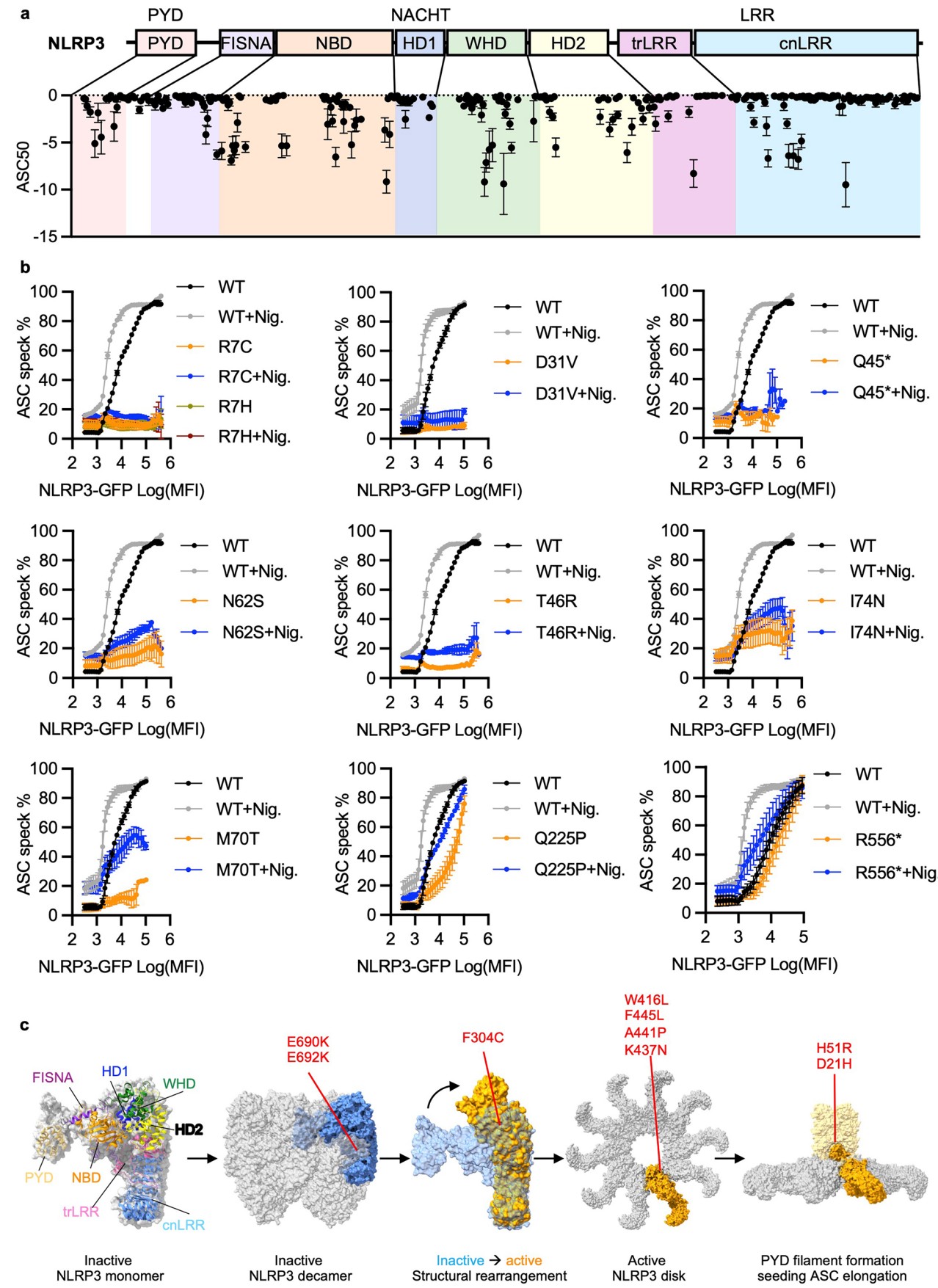

**Extended Data Fig. 5 | See next page for caption.**

**Extended Data Fig. 5 | Analysis of inactive NLRP3 variants with ASC50 and structural rearrangements during NLRP3 activation. a**. NLRP3 variants with ASC50 between -15 and 0. ASC-BFP HEK392T cells were analyzed using flow cytometry 12 hours post-transfection (bottom). The location of each variant is color-coded as in the schematic diagram of NLRP3 domain organization (above). PYD, pyrin domain; FISNA, fish-specific NACHT-associated domain; NBD, nucleotide-binding domain; HD, helical domain; WHD, winged helix domain; trLRR, transition leucine-rich repeat domain; cnLRR, canonical LRR domain. **b**. ASC speck percentage in the cell population with an increasing amount of NLRP3 expression for wild-type (WT), R7C, R7H, D31V, Q45*, N62S, T46R, I74N, M70T, Q225P and R556* NLRP3 in ASC-BFP HEK392T cells 12 hours post-transfection with or without 1-hour nigericin (Nig., 2 μM) treatment. The same WT control data is used for each variant, except for R556*, when results were generated from the same experiment. **c**. Schematic of NLRP3 in its inactive form (PDB:7PZC) showing the pyrin domain (PYD) in light orange, fish-specific NACHT-associated domain (FISNA) in purple, NACHT domain subdomains including nucleotide-binding domain (NBD) in orange, helical domain 1 (HD1) in blue, winged helix domain (WHD) in green, HD2 in yellow, transition leucine-rich repeat domain (trLRR) in pink and canonical LRR domain (cnLRR) in blue on the left. NLRP3 inactive monomers assemble into the inactive decamer which can be disrupted by variants such as E690K and E692K. In response to the activation signal, NLRP3 switches into the active form and the active NLRP3 conformation can be stabilized by variants such as F304C. Active NLRP3 then forms NLRP3 disk-like structure which can be promoted by variants such as W416L, F445L, A441P, K437N. The pyrin domain in the active NLRP3 disk forms filament, and the interaction between each monomer can be strengthen by variants such as H51R and D21H. Data were pooled from 3–12 independent repeats (**a**, *n* value for each variant is listed in Supplementary Table 1) or 3 independent repeats (**b**). Each dot represents a single variant (**a**, mean and s.e.m. in **a-b**).

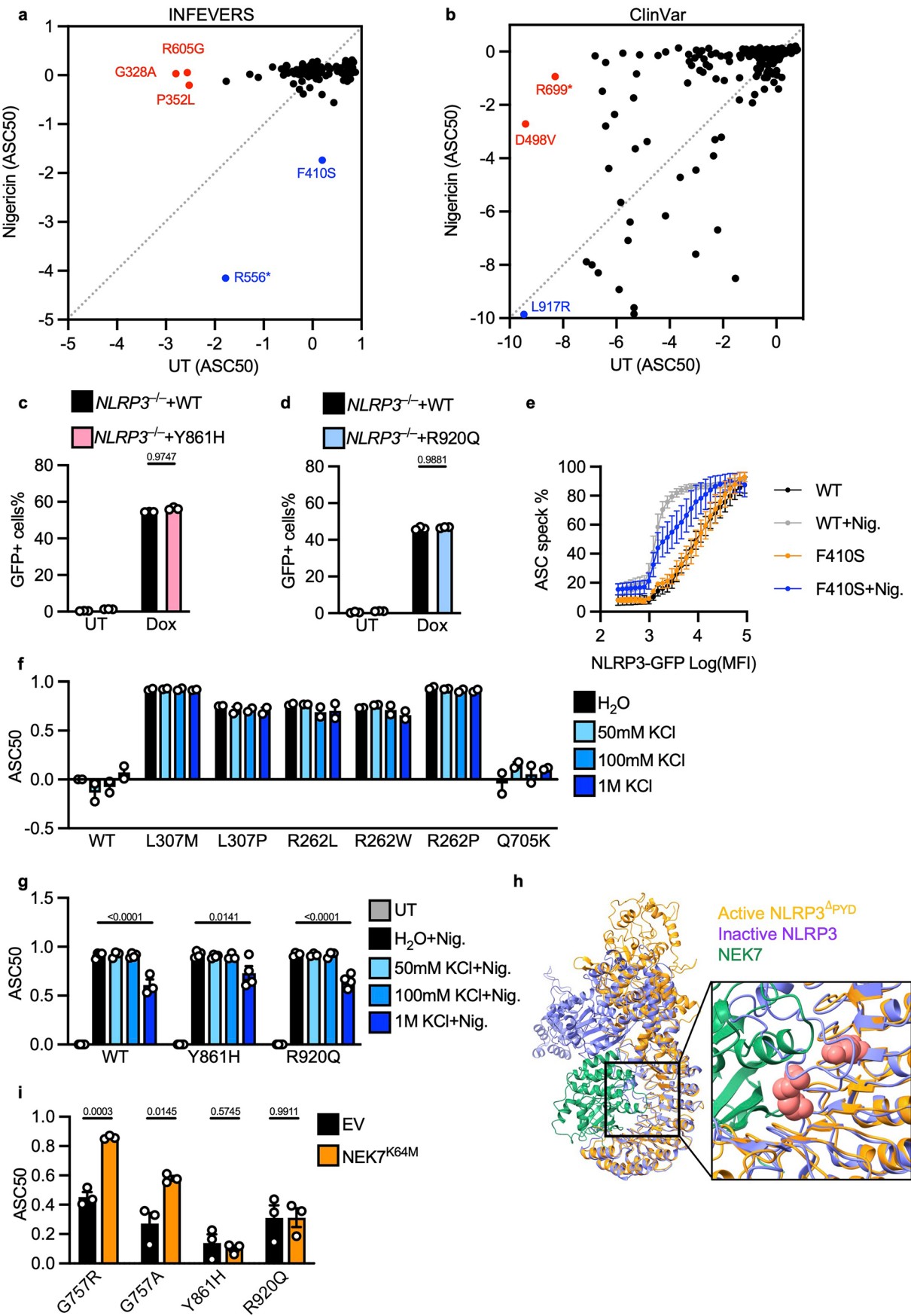

**Extended Data Fig. 6 | See next page for caption.**

**Extended Data Fig. 6 | Activity of NLRP3 variants in response to nigericin treatment. a-b**. Biplot for the ASC50 of untreated (UT) and nigericin (2 μM) treated NLRP3 variants from the INFEVERS database (**a**) or the ClinVar database (**b**). **c-d**. Percentage of cells expressing GFP-tagged wild-type (WT), Y861H (**c**), R920Q (**d**) in reconstituted THP-1 cells treated with or without doxycycline (Dox, 100 ng/mL) for 6 hours. **e**. ASC speck percentage in the cell population with an increasing amount of NLRP3 expression for wild-type (WT), and F410S NLRP3 in ASC-BFP HEK392T cells 12 hours post-transfection with or without 1-hour nigericin (Nig., 2 μM) treatment. The same WT control data is used as in Extended Data Fig. 5 for R556*, as results were generated from the same experiment. **f**. ASC50 of NLRP3 variants treated with H2O, 50 mM,100 mM, or 1 M KCl for 12 hours simultaneously with plasmids expression. **g**. ASC50 of NLRP3 variants pre-treated with $H_2O$, 50 mM,100 mM, or 1 M KCl, 30 mins prior to nigericin treatment. ASC50 is calculated in relative to the activity of each variant without nigericin treatment. **h**. Overlay of inactive NLRP3 in purple (PDB: 7PZC) with the NEK7-bound active NLRP3 in orange (PDB: 8EJ4). NEK7 is shown in green. Wild-type residues representing the location of Y861H and R920Q NLRP3 variants are highlighted as spheres in red. **i**. ASC50 for the NLRP3 variants overexpressed in ASC-BFP HEK293T cells with empty vector (EV) or mCherry-NEK7$^{K64M}$. Each dot represents a single variant (**a-b**), or one independent repeat (**c-d**, **f-g** and **i**). Data were pooled from 3–12 independent repeats (**a-b**, *n* value for each variant is listed in Supplementary Table 1), or 3 (**c-e**, **i**), 2 (**f**), 4 (**g**) independent repeats (mean in **a-b**, mean and s.e.m. in **c-g**, **i**). Two-tailed *t*-test in **c-d** and **i**, one-way ANOVA with Dunnett's multiple-comparisons test in **g**.

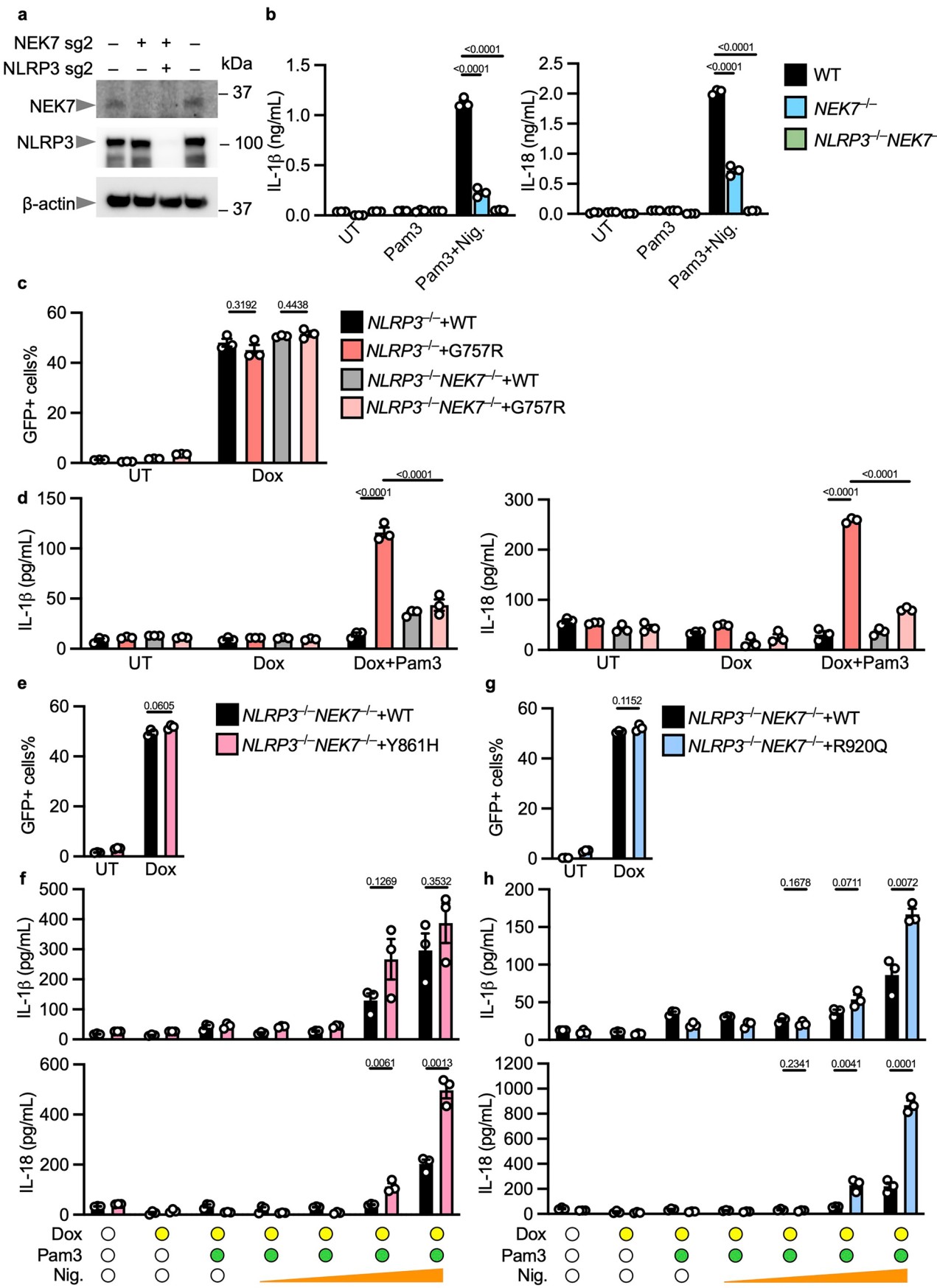

**Extended Data Fig. 7 | See next page for caption.**

**Extended Data Fig. 7 | NEK7 is not required for activity of nigericin-sensitive NLRP3 variants. a**. Immunoblot analysis of NEK7, NLRP3 and β-actin in WT THP-1 cells or CRISPR-knockout of NEK7 or NEK7 and NLRP3 together. **b**. Release of IL-1β and IL-18 from wild-type (WT), $NEK7^{-/-}$, $NLRP3^{-/-}NEK7^{-/-}$ THP-1 cells left untreated, or primed with Pam3CSK4 (Pam3, 100 ng/mL) for 14 hours and then with nigericin (Nig., 10 μM) for 1 hour. **c-d**. Percentage of cells expressing GFP (**c**) and the release of IL-1β and IL-18 (**d**) in $NLRP3^{-/-}$ or $NLRP3^{-/-}NEK7^{-/-}$ THP-1 cells reconstituted with GFP-tagged wild-type or G757R left untreated or primed with Pam3CSK4 (Pam3, 100 ng/mL) for 14 hours followed by 6 hours of doxycycline (Dox, 100 ng/mL) treatment. **e-h**. Percentage of cells expressing GFP and the release of IL-1β and IL-18 in $NLRP3^{-/-}NEK7^{-/-}$ THP-1 cells reconstituted with GFP-tagged wild-type (WT), Y861H (**e, f**) or R920Q (**g, h**), untreated or primed with Pam3CSK4 (Pam3, 100 ng/mL) for 14 hours followed by 6 hours of doxycycline (Dox, 100 ng/mL) treatment and increasing concentrations of nigericin (Nig., 0.5 μM, 1 μM, 5 μM, 10 μM) for 1 hour. Data was representative of 2 independent repeats (**a**), or pooled from 3 independent repeats (**b-h**). Each dot represents a single repeat (**b-h**, mean and s.e.m. in **b-h**). Two-tailed $t$-test in **c, e, g**, one-way ANOVA with Dunnett's multiple-comparisons test in **b, d, f, h**.

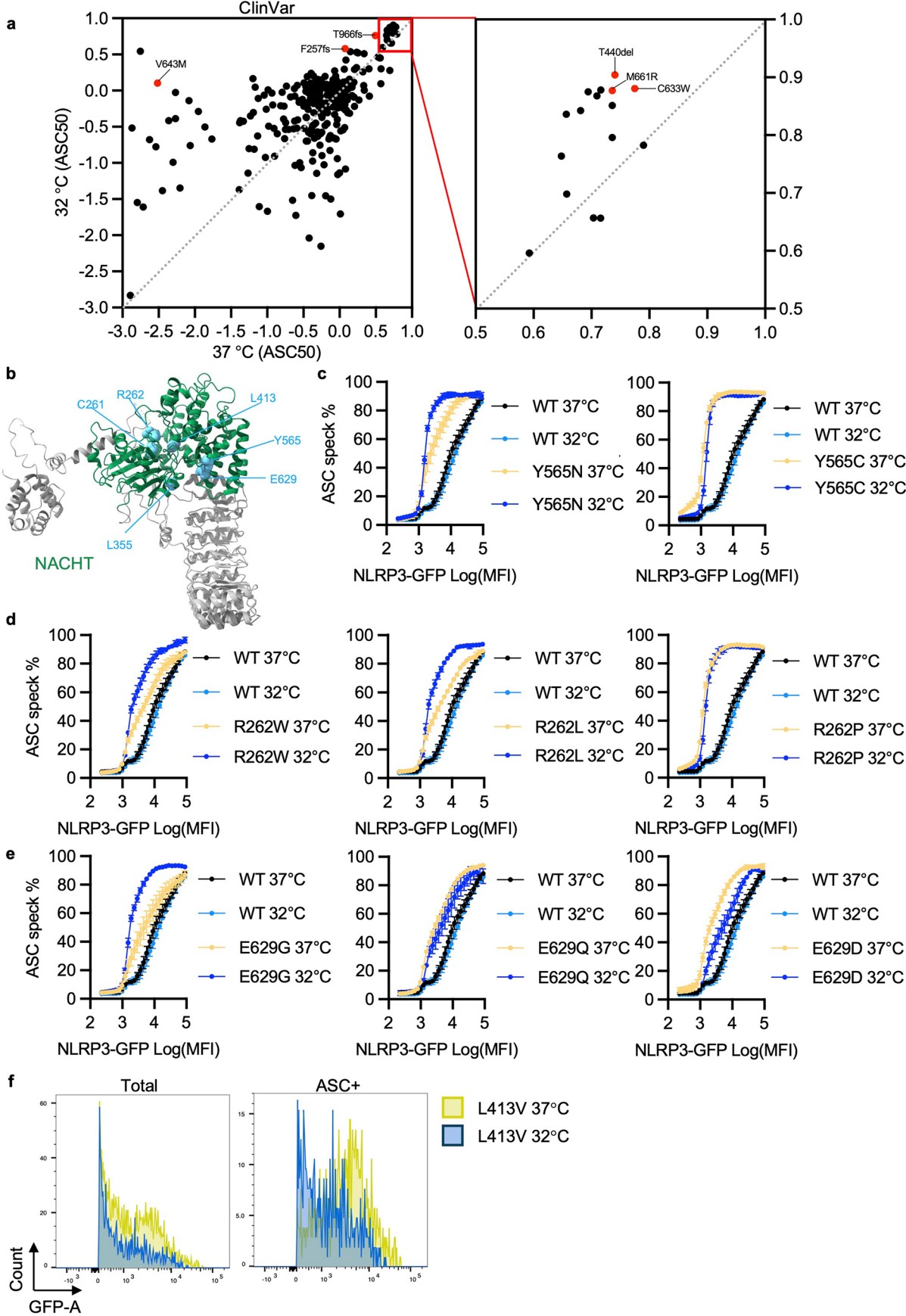

**Extended Data Fig. 8 | See next page for caption.**

**Extended Data Fig. 8 | Cold temperature exposure can activate specific NLRP3 variants. a**. Biplot overview (left) and inset (right) for ASC50 of NLRP3 variants (ClinVar database) incubated under 37- or 32-°C for 12 hours. Variants with significant increase of ASC50 are colored in red. **b**. Structural annotation of the location of disease-associated temperature-sensitive variants, highlighted in the blue sphere, in the inactive structure of NLRP3 (PBD: 7PZC). The NACHT domain is highlighted in green. **c-e**. ASC speck percentage in the cell population with increasing amount of NLRP3 expression for wild-type (WT), Y565N, and Y565C (**c**), R262W, R262L, and R262P (**d**) or E629G, E629Q, and E629D (**e**) NLRP3 variants incubated under 37- or 32-°C. **f**. Histogram of L413 expression under 37- or 32-°C analyzed using flow cytometry. The same WT control data is used when results were generated from the same experiment (**c-e**). Each dot represents a single variant (**a**). Data were pooled from 3–12 independent repeats (**a**, *n* value for each variant is listed in Supplementary Table 1) or 3 independent repeats (**c-e**, mean in **a**, mean and s.e.m. in **c-e**) or representative of 3 independent repeats (**f**).

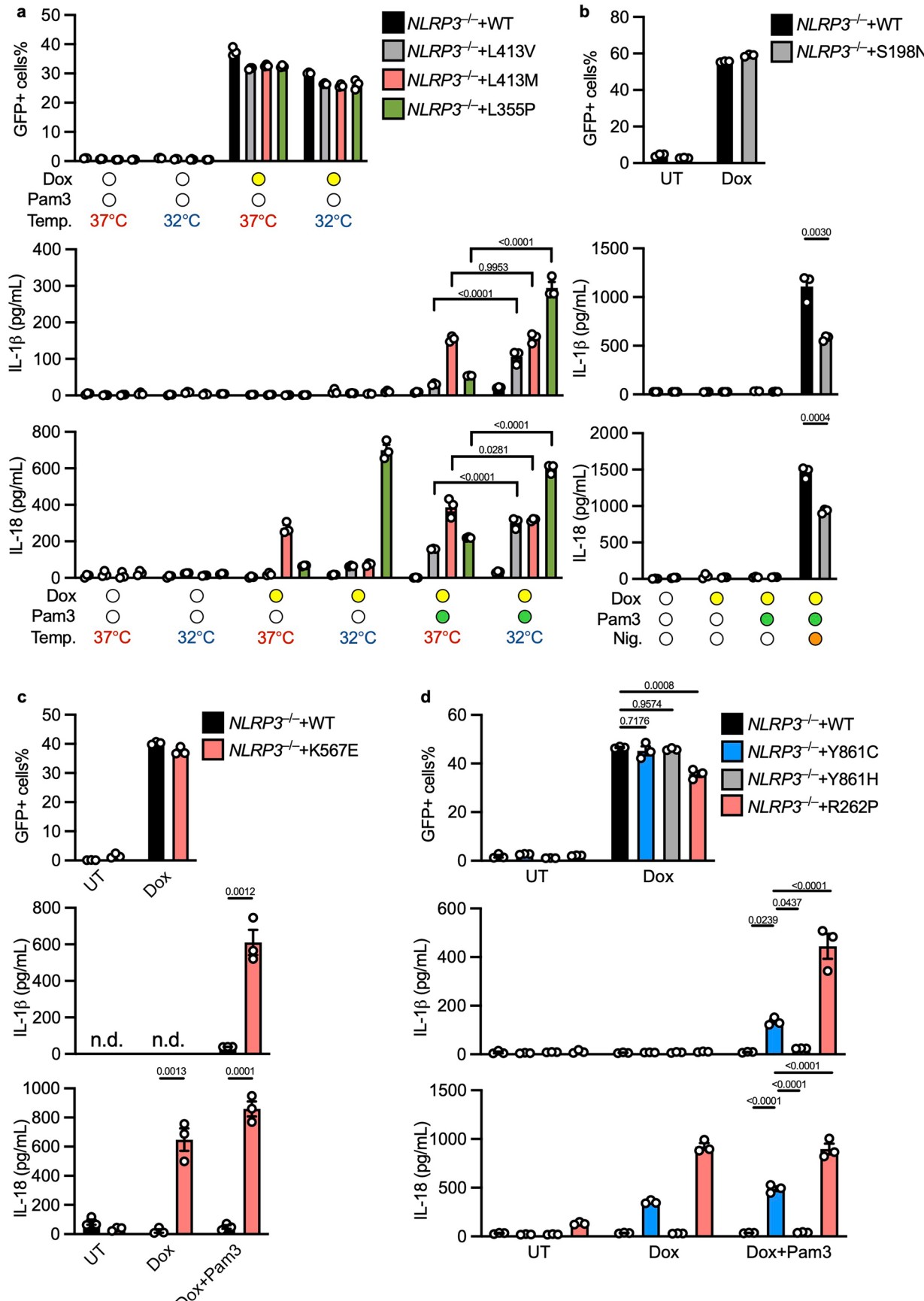

**Extended Data Fig. 9 | See next page for caption.**

**Extended Data Fig. 9 | Cold temperature exposure and post-translational modification contribute to NLRP3 activity in THP-1 cells. a**. Percentage of cells expressing GFP and the release of IL-1β and IL-18 in *NLRP3*^−/− THP-1 cells reconstituted with GFP-tagged wild-type (WT), L355P, L413M or L413V, untreated or primed with Pam3CSK4 (Pam3, 100 ng/mL) for 14 hours at 37 °C, followed by 6 hours of doxycycline (Dox, 50 ng/mL) treatment while incubating at 37 or 32 °C. **b**. Percentage of cells expressing GFP and the release of IL-1β and IL-18 in *NLRP3*^−/− THP-1 cells reconstituted with GFP-tagged wild-type (WT) or S198N left untreated or primed with Pam3CSK4 (Pam3, 100 ng/mL) for 14 hours followed by 6 hours of doxycycline (Dox, 100 ng/mL) and nigericin (Nig., 10 μM) for 1 hour. **c-d**. Percentage of cells expressing GFP and the release of IL-1β and IL-18 in *NLRP3*^−/− THP-1 cells reconstituted with GFP-tagged wild-type (WT), K567E (**c**), Y861C, Y861H or R262P (**d**) left untreated or primed with Pam3CSK4 (Pam3, 100 ng/mL) for 14 hours followed by 6 hours of doxycycline (Dox, 100 ng/mL). Each dot represents an independent repeat (**a-d**). Data were pooled from 3 independent repeats (**a-d**, mean and s.e.m. in **a-d**). One-way ANOVA with Dunnett's multiple-comparisons test in **a** and **d**, two-tailed *t*-test in **b-c**.

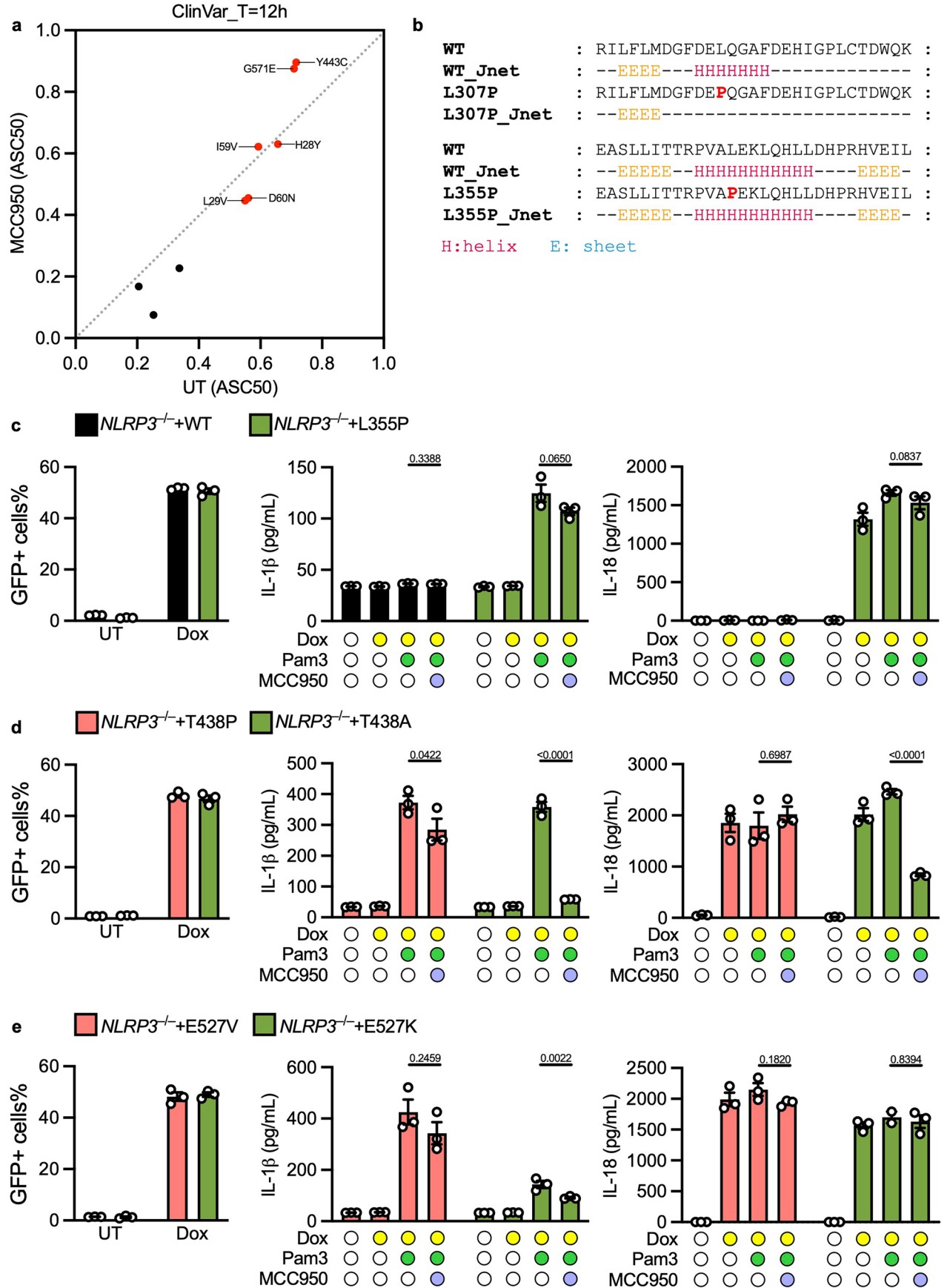

**Extended Data Fig. 10 | See next page for caption.**

**Extended Data Fig. 10 | Specific NLRP3 variants can resist inhibition by MCC950 in THP-1 cells. a**. Biplot for ASC50 of NLRP3 variants (ClinVar database) left untreated or treated with MCC950 (10 µM) for 12 hours while NLRP3 overexpression. **b**. Secondary structure prediction of wild-type (WT) and NLRP3 variants using Jpred 4. **c-e**. Percentage of cells expressing GFP and the release of IL-1β and IL-18 in *NLRP3*$^{-/-}$ THP-1 cells reconstituted with GFP-tagged wild-type (WT), L355P (**c**), T438P or T438A (**d**), E527V or E527K (**e**) left untreated or primed with Pam3CSK4 (Pam3, 100 ng/mL) for 14 hours followed by 6 hours of doxycycline (Dox, 100 ng/mL) together with or without MCC950 (10 µM). Each dot represents one variant (**a**) or one independent repeat (**c-e**). Data were pooled from 3–12 independent repeats (**a**, *n* value for each variant is listed in Supplementary Table 1) or 2-3 independent repeats (**c-e**, mean in **a**, mean and s.e.m. in **c-e**). One-way ANOVA with Dunnett's multiple-comparisons test in in **c-e**.

# Reporting Summary

## Statistics

For all statistical analyses, confirm that the following items are present in the figure legend, table legend, main text, or Methods section.

| n/a | Confirmed | |
|---|---|---|
| ☐ | ☒ | The exact sample size (*n*) for each experimental group/condition, given as a discrete number and unit of measurement |
| ☐ | ☒ | A statement on whether measurements were taken from distinct samples or whether the same sample was measured repeatedly |
| ☐ | ☒ | The statistical test(s) used AND whether they are one- or two-sided <br> *Only common tests should be described solely by name; describe more complex techniques in the Methods section.* |
| ☒ | ☐ | A description of all covariates tested |
| ☒ | ☐ | A description of any assumptions or corrections, such as tests of normality and adjustment for multiple comparisons |
| ☐ | ☒ | A full description of the statistical parameters including central tendency (e.g. means) or other basic estimates (e.g. regression coefficient) AND variation (e.g. standard deviation) or associated estimates of uncertainty (e.g. confidence intervals) |
| ☐ | ☒ | For null hypothesis testing, the test statistic (e.g. *F*, *t*, *r*) with confidence intervals, effect sizes, degrees of freedom and *P* value noted <br> *Give P values as exact values whenever suitable.* |
| ☒ | ☐ | For Bayesian analysis, information on the choice of priors and Markov chain Monte Carlo settings |
| ☒ | ☐ | For hierarchical and complex designs, identification of the appropriate level for tests and full reporting of outcomes |
| ☐ | ☒ | Estimates of effect sizes (e.g. Cohen's *d*, Pearson's *r*), indicating how they were calculated |

*Our web collection on statistics for biologists contains articles on many of the points above.*

## Software and code

Policy information about availability of computer code

| Data collection | FACS: BD FACSDiva™ Software (v9.0); SpectroFlo (v3.2.1) <br> ELSA: MARS (V3.01 R2) <br> Imaging flow cytometry: IDEAS software (V6.2) |
|---|---|
| Data analysis | GraphPad Prism (version 9.5.1) <br> Flowjo (10.8.2) <br> R Studio (version 2024.04.2+764) <br> flowCore (release no. bioc_3.13, https://github.com/RGLab/flowCore) <br> flowAI (release no. flowise@1.4.0, https://github.com/FlowiseAI/Flowise) <br> openCyto (release no. bioc_3.13, https://github.com/RGLab/openCyto) <br> drda (Malyutina, A. et al 2023, https://github.com/albertopessia/drda) <br> data2bfactor (2003 Robert L. Campbell, https://peterslab.org/downloads.php) <br> Autospeck workflow (this study, https://github.com/Seth-Masters-Lab/Speck-Assay-Automated-Workflow.git) <br> ChimeraX (version 1.7) <br> Pymol (version 2.5.0, Schrödinger, LLC) <br> Dynamut (Rodrigues, C. H., et al. 2018) <br> DDmut (Zhou, Y., et al. 2023) <br> mCSM-lig (Pires, D. E., et al. 2016) <br> Jpred (v4, Cole, C., 2008) <br> ImageLab Software (v6.01) |

For manuscripts utilizing custom algorithms or software that are central to the research but not yet described in published literature, software must be made available to editors and reviewers. We strongly encourage code deposition in a community repository (e.g. GitHub). See our guidelines on research reproducibility and openness.

```
Quanstudio Design and Analysis Software  (v2.8)
ChatGPT (v4)
```

For manuscripts utilizing custom algorithms or software that are central to the research but not yet described in published literature, software must be made available to editors and reviewers. We strongly encourage code deposition in a community repository (e.g. GitHub). See the Nature Portfolio guidelines for submitting code & software for further information.

## Data

Policy information about availability of data

All manuscripts must include a data availability statement. This statement should provide the following information, where applicable:
- Accession codes, unique identifiers, or web links for publicly available datasets
- A description of any restrictions on data availability
- For clinical datasets or third party data, please ensure that the statement adheres to our policy

Our data is deposited in publicly available data repositories including INFEVERS (identifier: NLRP3, https://infevers.umai-montpellier.fr) and GenIA (identifier: NLRP3, https://www.geniadb.net/app/ref/info.php?id=1380). Databases used in this study include: INFEVERS (identifier: NLRP3, https://infevers.umai-montpellier.fr/web/), ClinVar (identifier: NLRP3, https://www.ncbi.nlm.nih.gov/clinvar/), AlphaMissense (identifier: NLRP3, https://alphamissense.hegelab.org/search), gnomAD (v4.1.0, identifier: NLRP3, https://gnomad.broadinstitute.org/gene/ENSG00000162711?dataset=gnomad_r4). Datasets used in this study are available in the RCSB Protein Data Bank (https://www.rcsb.org ) with accession-codes including PDB:8EJ4, 7PZC, 6NPY and 7PZD.

## Research involving human participants, their data, or biological material

Policy information about studies with human participants or human data. See also policy information about sex, gender (identity/presentation), and sexual orientation and race, ethnicity and racism.

| | |
|---|---|
| Reporting on sex and gender | Not applicable. |
| Reporting on race, ethnicity, or other socially relevant groupings | Not applicable. |
| Population characteristics | Not applicable. |
| Recruitment | Not applicable. |
| Ethics oversight | Human peripheral blood mononuclear cells (PBMCs) were sourced from the Victorian Blood Donor Registry under WEHI HREC approval 18/07. |

Note that full information on the approval of the study protocol must also be provided in the manuscript.

# Field-specific reporting

Please select the one below that is the best fit for your research. If you are not sure, read the appropriate sections before making your selection.

☒ Life sciences  ☐ Behavioural & social sciences  ☐ Ecological, evolutionary & environmental sciences

For a reference copy of the document with all sections, see nature.com/documents/nr-reporting-summary-flat.pdf

# Life sciences study design

All studies must disclose on these points even when the disclosure is negative.

| | |
|---|---|
| Sample size | Sample size, or the number of NLRP3 variants tested, id determined based on availability. NLRP3 variants tested in this study were sourced from publicly available datasets including INFEVERS and ClinVar. Additional variants were sourced from clinician in contact. |
| Data exclusions | FACS data were processed in a fully automated manner using the R code generated in this study. Sample with poor quality where the dose-response curve fitting failed to be generated by the code were excluded or labelled as out of scale where appropriate. |
| Replication | All experiments were independently repeated at least 2 times as indicated in figure legends. Where possible, findings were confirmed in different experimental systems. |
| Randomization | Transfected or reconstituted cells used in this study represent pre-defined experimental groups. These were created to test specific hypotheses regarding gene function and variant effects, making randomization unnecessary. |
| Blinding | Controls were used in each experimental conditions, and results were recorded automatically or analyzed through software and thus reducing the potential for subjective influence . Samples or cell lines were coded during experiments providing partial blinding measurement. |

# Reporting for specific materials, systems and methods

We require information from authors about some types of materials, experimental systems and methods used in many studies. Here, indicate whether each material, system or method listed is relevant to your study. If you are not sure if a list item applies to your research, read the appropriate section before selecting a response.

## Materials & experimental systems

| n/a | Involved in the study |
|-----|----------------------|
| ☐ | ☒ Antibodies |
| ☐ | ☒ Eukaryotic cell lines |
| ☒ | ☐ Palaeontology and archaeology |
| ☒ | ☐ Animals and other organisms |
| ☒ | ☐ Clinical data |
| ☒ | ☐ Dual use research of concern |
| ☒ | ☐ Plants |

## Methods

| n/a | Involved in the study |
|-----|----------------------|
| ☒ | ☐ ChIP-seq |
| ☐ | ☒ Flow cytometry |
| ☒ | ☐ MRI-based neuroimaging |

## Antibodies

| | |
|---|---|
| Antibodies used | NLRP3 (1:1000, rabbit, CST, 15101S)<br>NEK7 (1:1000,rabbit, CST, 3057S)<br>Tubulin (1:5000, rat, Santa Cruz Biotechnology, sc-53029)<br>β-actin conjugated with horseradish peroxidase (1:5000, Santa Cruz Biotechnology, SANTSC-47778HRP)<br>Goat anti-rat horseradish peroxidase-conjugated secondary antibodies (1:5000, Invitrogen, 31470)<br>Goat anti-Rabbit IgG (H+L) Secondary Antibody, HRP (1:5000, Invitrogen, 31460) |
| Validation | All antibodies were validated by the manufacturers and confirmed using knockout cells in this study (NLRP3, NEK7). Links for additional validation report:<br>NLRP3 (rabbit, CST, 15101S)<br> https://www.cellsignal.com/products/primary-antibodies/nlrp3-d4d8t-rabbit-mab/15101?srsltid=AfmBOorLKaVqqtz6o42ayCbJ8PT2ACME9jnPpFG38gvUeTdmer6w3Qtf<br>NEK7 (rabbit, CST, 3057S)<br>https://www.cellsignal.com/products/primary-antibodies/nek7-c34c3-rabbit-mab/3057?srsltid=AfmBOoq62uRPLP1WQ7P-3RA3Vl6qoZyc3_fTQMfZCrrmeD8mVh2C6Uik<br>Tubulin (rat, Santa Cruz Biotechnology, sc-53029)<br>https://www.scbt.com/p/alpha-tubulin-antibody-yl1-2?srsltid=AfmBOopMr1MgMVp83IFt3ucQ55hXaGc4c7mRLsdK8asbf5YBo69ybwxZ<br>β-actin-HRP (Santa Cruz Biotechnology, SANTSC-47778HRP)<br>https://www.scbt.com/p/beta-actin-antibody-c4?srsltid=AfmBOooldZZeATWlAMBYB5mVcMGlAikozpoIA7kcbJHIQ0y7k-NKM_v-<br>anti-rat horseradish peroxidase-conjugated secondary antibodies (Invitrogen, 31470)<br>https://www.thermofisher.com/antibody/product/Goat-anti-Rat-IgG-H-L-Secondary-Antibody-Polyclonal/31470<br>Goat anti-Rabbit IgG (H+L) Secondary Antibody, HRP (Invitrogen, 31460)<br>https://www.thermofisher.com/antibody/product/Goat-anti-Rabbit-IgG-H-L-Secondary-Antibody-Polyclonal/31460 |

## Eukaryotic cell lines

Policy information about cell lines and Sex and Gender in Research

| | |
|---|---|
| Cell line source(s) | Human embryonic kidney (HEK) 293T cells (ATCC CRL-3216)<br>THP-1 cells (ATCC TIB-202)<br>U937 (ATCC CRL-1593.2) |
| Authentication | All cell lines were authenticated using ATCC's human cell authentication service utilizing short tandem repeat (STR) profiling. |
| Mycoplasma contamination | All cell lines were confirmed negative for mycoplasma by PCR or MycoStrip (InvivoGen, rep-mysnc-100) throughout the duration of this study. |
| Commonly misidentified lines<br>(See ICLAC register) | No commonly misidentified cell lines were used. |

# Plants

| | |
|---|---|
| Seed stocks | *Report on the source of all seed stocks or other plant material used. If applicable, state the seed stock centre and catalogue number. If plant specimens were collected from the field, describe the collection location, date and sampling procedures.* |
| Novel plant genotypes | *Describe the methods by which all novel plant genotypes were produced. This includes those generated by transgenic approaches, gene editing, chemical/radiation-based mutagenesis and hybridization. For transgenic lines, describe the transformation method, the number of independent lines analyzed and the generation upon which experiments were performed. For gene-edited lines, describe the editor used, the endogenous sequence targeted for editing, the targeting guide RNA sequence (if applicable) and how the editor was applied.* |
| Authentication | *Describe any authentication procedures for each seed stock used or novel genotype generated. Describe any experiments used to assess the effect of a mutation and, where applicable, how potential secondary effects (e.g. second site T-DNA insertions, mosiacism, off-target gene editing) were examined.* |

# Flow Cytometry

## Plots

Confirm that:

☒ The axis labels state the marker and fluorochrome used (e.g. CD4-FITC).

☒ The axis scales are clearly visible. Include numbers along axes only for bottom left plot of group (a 'group' is an analysis of identical markers).

☒ All plots are contour plots with outliers or pseudocolor plots.

☒ A numerical value for number of cells or percentage (with statistics) is provided.

## Methodology

| | |
|---|---|
| Sample preparation | For flow cytometry, HEK293T cells stably expressing BFP-ASC were transfected with plasmids expressing GFP-NLRP3. Cells were collected and fixed in pre-chilled methanol or in pre-warmed 4% PFA and resuspended in PBS supplemented with 2% FCS and 5 mM EDTA. THP-1 cells were directly resuspended in PBS supplemented with 2% FCS and 5 mM EDTA. |
| Instrument | Amnis ImageStreamX MKII imaging flow cytometer<br>BD LSR Fortessa X20<br>Cytek Aurora |
| Software | IDEAS software (V6.2)<br>BD FACSDiva™ Software (v9.0)<br>SpectroFlo (v3.2.1) |
| Cell population abundance | At least 10,000 cells were acquired for each sample. |
| Gating strategy | Samples were gated for non-debri single cells based on FSC/SSC and then gated on ASC speck based on BFP-A/BFP-W and GFP with GFP-A/FSC-A. |

☒ Tick this box to confirm that a figure exemplifying the gating strategy is provided in the Supplementary Information.

