## [Peer Review File · Nature Immunology]

Mechanisms of NLRP3 activation and inhibition elucidated by functional analysis of disease-associated variants.

Corresponding Author: Dr Seth Masters

Version 0:

Decision Letter:

27th Mar 2024

Dear Dr Masters,

Your Resource, "Mechanisms of NLRP3 activation and inhibition elucidated by functional analysis of disease-associated variants." has now been seen by 2 referees. You will see from their comments copied below that while they find your work of considerable potential interest, they have raised quite substantial concerns that must be addressed. In light of these comments, we cannot accept the manuscript for publication, but would be very interested in considering a revised version that addresses these serious concerns.

We hope you will find the referees' comments useful as you decide how to proceed. If you wish to submit a substantially revised manuscript, please bear in mind that we will be reluctant to approach the referees again in the absence of major revisions.

We encourage you to address all of the reviewers concerns, in particular those of Reviewer #2 concerning validation of results in primary cells.

If you choose to revise your manuscript taking into account all reviewer and editor comments, please highlight all changes in the manuscript text file [OPTIONAL: in Microsoft Word format].

* If you have not done so already please begin to revise your manuscript so that it conforms to our Resource format instructions at <http://www.nature.com/ni/authors/index.html>. Refer also to any guidelines provided in this letter.

The Reporting Summary can be found here:
<https://www.nature.com/documents/nr-reporting-summary.pdf>

When submitting the revised version of your manuscript, please pay close attention to our <https://www.nature.com/nature-portfolio/editorial-policies/image-integrity> Digital Image Integrity Guidelines. and to the following points below:

Finally, please ensure that you retain unprocessed data and metadata files after publication, ideally archiving data in

perpetuity, as these may be requested during the peer review and production process or after publication if any issues arise.

Link Redacted

If you wish to submit a suitably revised manuscript we would hope to receive it within 6 months. If you cannot send it within this time, please let us know. We will be happy to consider your revision so long as nothing similar has been accepted for publication at Nature Immunology or published elsewhere.

Nature Immunology is committed to improving transparency in authorship. As part of our efforts in this direction, we are now requesting that all authors identified as 'corresponding author' on published papers create and link their Open Researcher and Contributor Identifier (ORCID) with their account on the Manuscript Tracking System (MTS), prior to acceptance. ORCID helps the scientific community achieve unambiguous attribution of all scholarly contributions. You can create and link your ORCID from the home page of the MTS by clicking on 'Modify my Springer Nature account'. For more information please visit www.springernature.com/orcid.

Thank you for the opportunity to review your work.

Sincerely,

Stephanie Houston, PhD
Senior Editor
Nature Immunology

Reviewers' Comments:

Reviewer #1:

Remarks to the Author:

In their clearly presented manuscript, "Mechanisms of NLRP3 activation and inhibition elucidated by functional analysis of disease-associated variants.", Feng et al. take a methodical and comprehensive approach to characterize the functional significance of all described NLRP3 variants and study the mechanisms of specific CAPS mutations. They use recombinant HEK cells, flow cytometry, structural modeling, as well as ion changes, cold temperature, and pharmacologic approaches to produce a long awaited and clinically useful resource for clinicians and scientists.

The strengths of this paper are

1. Logical evaluation of all reported NLRP3 variants (including common low penetrance variants and variants in other domains besides NACHT) that explores several mechanisms including cold and MCC950 responses that is relevant to CAPS patients.
2. Detailed structural modeling to visualize molecular function.
3. Authors do a nice job of not overstating and recognizing importance of clinical presentation.
4. Data and interpretation that come together for an understandable story.

The primary weaknesses

1. More discussion about the clinical CAPS spectrum as an explanation why some mutations are less straightforward
2. While I really like the structure figures, I am concerned that some readers not familiar with inflammasome structure and function may have difficulty interpreting them. It might be helpful to include some supplemental cartoon figures to illustrate the points they are trying to get across.

Comments:

Title: Accurate and succinct"

Abstract: Well written and concise

Introduction

Concise and comprehensive and appropriate for this audience
Good review of current state of knowledge of this disease and mechanisms

Methods

Adequate detail of techniques

Results

Clear presentation of data

Figure legends adequately detailed

Minor points –

The authors mention L307P (with L355P) as a common FCAS mutation but do not show any cold data. I don't understand vibrational entropy.

Supplementary Figures

Appropriate placement of supplemental data.

Discussion

Data supports conclusions

References - Adequate

Reviewer #2:

Remarks to the Author:

In this manuscript, Feng et al. adapted the Time of Flight for inflammasome Evaluation (TOFIE) assay in HEK293T cells and single cell analysis by flow cytometry to investigate gain-of-function of >500 natural variants of NLRP3 found in the Infevers and ClinVar databases. Although the assay has been already described by others since 2016 (PMID: 27479658), the originality of this manuscript relies on the standardized statistical analysis and the inclusion of by very far the largest numbers of NLRP3 variants in a comparative study. Results of these analysis identify gain-of-function mutations in this assay and therefore provide an important resource for CAPS diagnosis. CAPS is often efficiently treated with anti-IL1, and diagnosis is critical but pathogenicity of most NLRP3 variants is poorly characterized. Therefore, these results will certainly have immediate impact on the patients care. In addition, identification of the key residues for the control of NLRP3 activity provides also new insights on NLRP3 regulation mechanism. Nevertheless, the major weakness of this manuscript is that results rely only on one experimental system (overexpression in HEK293T) which is very artificial and highly prone to artifacts, and results are not confirmed by physiological models other than the patient symptoms for some picked mutations. As it is a rare disease with very small cohort for each mutation, and that the phenotype/genotype correlation is not strong (lots of variability between patients of the same family, some patients show symptoms from different disease entities FCAS, MWS, NOMID, which consist more of a spectrum than distinct clinical entities), the validation of the results is questionable. Therefore, although the conclusion of the manuscript may lead to major advances and impact for both the CAPS and the inflammasome fields, the manuscript requires major revisions to be accepted in Nature Immunology.

Major points :

- In myeloid cells expressing endogenous NLRP3, NLRP3 is regulated by translational and post-translational modifications upon priming and activation signals. The TOFIE assay in HEK293T cells detects intrinsic NLRP3 gain-of-functions. Results in table 1 are based on untreated cells, therefore only constitutive active mutants can be detected, ignoring mutants that could show a lower activation threshold to priming or activation signals. In table 2, only a small selection of mutants (116/529) is investigated in response to Nigericin as an activation signal and cold temperature. The role of priming is mostly considered by the authors through the aspect of NLRP3 up-regulation and assessed by measuring NLRP3-GFP fluorescence level. Nevertheless, priming is now recognized to include complex, and very poorly understood, post-translational modifications controlling NLRP3 conformational changes, interaction with partners and subcellular relocalization (PMID: 32745339). Overexpression in HEK293T cells may bypass most of these regulations, and therefore possible gain-of-function mutants may be missed.

- A major limitation of the study is the use of only one read-out (ASC speck) in a very artificial system. Noteworthy, in human monocytes, NLRP3 has been shown to form alternative inflammasome without ASC speck (PMID: 27037191). Do PBMCs from CAPS patients show speck spontaneously or upon LPS priming (in conditions where pyroptosis or IL-1b are detected)? It would be informative to compare (1) mutants with high ASC50 and (2) others with low ASC50 in untreated conditions but with higher sensitivity to Nig or cold temperature, and finally (3) mutants without any detectable increased IC50 in any of the conditions, for speck formation in PBMCs ex vivo.

- Tables 1-3: It would be informative to add as additional columns for each mutation the associated symptom(s) (FCAS, MWS, NOMID) and whether each mutation has been identified as symptomatic when somatic or/and germline. While authors referred to patient symptoms as the main confirmation method for their data in HEK293T-based assay, it often relies only on few picked examples. Systematic analysis would be more convincing.

- Fig 1b: Some patients have some phenotypes typical to different historical disease entities (FCAS, MWS, NOMID), and some variants are found in patients with different disease entities that correspond more to a spectrum of one disease. The authors should indicate the criteria used to link each variant to the corresponding FCAS, MWS, NOMID groups. For example, R262W is associated with FACS and MWS in the Infevers database. What are the criteria to associate R262W to FACS rather than MWS in Fig 1b? In addition, to be more convincing, all variants for one given site should be included in the figure. For example, R262Q could be added. Additional examples, other than R262, A441 and 629 could also be added in

fig1b or as supplemental figures.

- A subset of 116 variants among the 529 variants included in the study has been investigated in response to nigericin treatment. The criteria to choose these 116 variants are unclear as many "unknown significance" variants (table 1 column E) were not included in table 2. For the analysis to be comprehensive, all variants without constitutive activity detected in table 1 may be tested in response to nigericin signal. In addition, other activation signals could be tested including particles and K⁺-independent stimuli (Imiquimod-derivatives), as mechanisms and regulatory structure may differ (as does nigericin- and temperature-dependent activation mechanism)
- In an attempt to validate the experimental system, the authors should compare the expression level of NLRP3 and ASC in the HEK293T vs primary myeloid cells of, at least, some human cell lines.
- Line 247-253: NEK7 has been shown to be dispensable in several human cells for inflammasome activation. Does HEK293T cells used in this study express NEK7 and is it required for ASC oligomerization in the ASC50 assay? Use of NEK7 KO HEK293T would definitely rule out the impact of NEK7-NLRP3 interaction in the activity of the LRR mutants.
- Is temperature-induced activity in HEK293T cells a common feature of all FCAS mutations, and a discriminant parameter between the FCAS vs MWS or NOMID. To correlate FCAS and cold-induced activation in HEK293T cells, it may be informative to distinguish FACS and CAPS undefined mutations in table 2 and in Fig4A. In addition, it would be informative to test cold-induced activation in HEK293T cells for all well characterized MWS and NOMID, as it has been done with FCAS.

Minor points

- Line 54 : "ATP hydrolysis which is important to induce structural rearrangement and inflammasome assembly". This sentence is confusing and should be modified to clearly indicate that ATP hydrolysis is important to switch from the active to the inactive form and not the reverse (PMID: 35114687).
- Line 57: transcription-independent priming has also been described (PMID: 22948162)
- Line 97: the reference PMID: 36562561 is missing. In addition a preprint showed recently that primary monocytes from patients bearing NLRP3 E525K are resistant to MCC950 and could be cited as well (doi:<https://doi.org/10.1101/2023.09.22.558949>)
- Supplementary fig 1C: this is not very clear how the middle and the right panels relate one to another. It may help to indicate the ASC50% in the middle panel. In addition, the choice of identical color (blue and red) for different things (Speck⁻/speck⁺ vs mutant/WT) is also confusing.
- Line 142: to increase clarity, the authors may indicate that positive ASC50 corresponds to NLRP3 hyperactivity. Apparently, these mutants are distributed all across the NACHT domain and not concentrated in the NBD and HD2 subdomains as stated by the authors. Related to this last point, the authors may indicate how this "concentration" was assessed (method to calculate the frequency values?).
- table 1: The authors should indicate what means "out of scale".
- Table 2: It would be helpful if the author can provide a cutoff to identify mutants hypersensitive to nigericin activation signal. For example, authors conclude that Q705K (Nig ASC50=0.116) is not pathogenic, but that R920Q, Y861H, Y861C are hypersensitive to Nig (Nig ASC50=0.128, 0.146, 0.169 respectively), while Nig ASC50 are very similar.
- tables 1-3: It may help to add the NLRP3 domain corresponding to each mutation. In addition, the same order should be used for different substitutions of one aa (Y861H and Y861C are inverted between table 1 and 2)
- It would be also useful to add in table 2, the data obtained in untreated condition (table 1) to compare more easily the results of the different experimental conditions for one given variant on one single sheet. In addition, graphical representation of the data would be useful (using Fig1a as a model)
- This is confusing that the authors use in this manuscript a different nomenclature than in the Infevers database to label the mutations (resulting in a 2 aa shift). In addition, in Line 393-396 the Infevers nomenclature is used, inconsistently with the rest of the manuscript.
- Line 235: "KCl did inhibit nigericin-induced activation for all". According to table 2, not all mutants with increased sensitivity to Nig were tested in presence of Nig and increased KCl. What are the criteria used to pick the 5 mutants tested for sensitivity to KCl?
- Line 242: this conclusion is very similar to the conclusion of a recent preprint (Cosson et al. ref 51), which may be cited here.
- Most mutations symptomatic when somatic are not found as germline, suggesting that mutations with strong gain of functions may be lethal as germline (PMID: 31816408). Therefore symptomatic somatic mutations may correspond to mutation with strong gain-of-function. Correlation between ASC50 score and germline vs somatic symptomatic mutations may be informative.
- The manuscript relies exclusively on highly processed data. Additional FACS plots (for each NLRP3-GFP increment gatings), for WT and few mutants would be informative for the reader. In addition, microscopy images of FACS sorted Speck⁺ and Speck⁻ cells to control the ASC specks are necessary.
- Fig5a,b : some mutations are activated by MCC950 treatment (for example at 12h, a dot with MCC950(ASC50) around 0.42 and UT(ASC50) around 0.1). It is surprising that MCC950 may be able to bind the active conformation of NLRP3. The authors should comment on that and confirm their data in myeloid cells competent of NLRP3 activation.
- Line 380-383: It should be added here that in addition to the upregulation of NLRP3 expression, priming results in NLRP3 post-translational modifications and relocalization that render NLRP3 responsive to activation signal (PMID: 32745339).
- Line 382-383: to validate this hypothesis, authors should test whether (1) variants activated by nigericin (table 2), and (2) variants hyper-responsive to signal 2 in the cited preprint (ref 51) show higher NLRP3 expression level.
- Line 393-396: The gain-of-function of Q705K is highly discussed including in functional assay (PMID: 22529966), and although it is a frequent variant in the healthy individuals, its prevalence is increased in autoinflammatory patients (PMID: 32477355). This suggests that this mutant is a gain-of-function pathogenic mutant although it may not be sufficient to trigger the disease depending on the environmental exposome and other genetic factors.
- Different fixation method (methanol or PFA) are used depending on the cell treatments. The authors should explain why

different protocol were used and provide controls to assess the stability of ASC50 values following different fixation procedures. Can the values in the different tables be compared?

Version 1:

Decision Letter:

Our ref: NI-RS37501A

5th Dec 2024

Dear Dr. Masters,

Thank you for submitting your revised manuscript "Mechanisms of NLRP3 activation and inhibition elucidated by functional analysis of disease-associated variants." (NI-RS37501A). It has now been seen by the original referees and their comments are below. The reviewers find that the paper has improved in revision, and therefore we'll be happy in principle to publish it in Nature Immunology, pending minor revisions to satisfy the referees' final requests and to comply with our editorial and formatting guidelines.

We will now perform detailed checks on your paper and will send you a checklist detailing our editorial and formatting requirements in about a week. Please do not upload the final materials and make any revisions until you receive this additional information from us.

If you had not uploaded a Word file for the current version of the manuscript, we will need one before beginning the editing process; please email that to immunology@us.nature.com at your earliest convenience.

Thank you again for your interest in Nature Immunology Please do not hesitate to contact me if you have any questions.

Sincerely,

Stephanie Houston, PhD
Senior Editor
Nature Immunology

Reviewer #2 (Remarks to the Author):

The authors replied adequately to most points. Their effort to validate some of the results in myeloid cell lines is highly appreciated. The last remaining point concerns the description of the statistical analysis, that I think should be more detailed to ease the understanding by the reader (see below). I also raised few minor points to be edited, to better reflect the literature.

Once the authors would have addressed these points, the manuscript would be suitable for publication in Nature Immunology. This comprehensive analysis of all NLRP3 natural variants will be a highly valuable resources for CAPS diagnosis and the understanding of NLRP3 activation and regulation mechanisms.

Major points:

- Statistical analysis. To my opinion, the statistical criteria to determine which mutant may be auto-activated, hypersensitive to Nig or activated by low temp, are still unclear. The method to calculate p values should be added to the methods section. In tables 2 and 3, it would be helpful to add which values are compared in the headline of the p-value column (as done in table 4). This is unclear whether the compared values are: (1) Nig Vs UT for a given variant or (2) Nig.variant vs Nig.WT. Adding p-values for UT ASC50 for each given variant compared to WT NLRP3 in table 1 should be informative as well. I also could not find the number of independent repeats.

Minor points:

- Line 71: Priming controls NLRP3 not only by its deubiquitination, but also many other modifications including -but not limited to- JNK1-dependent phosphorylation (PMID: 28943315). This line may be modified to reflect all the complexity of the NLRP3 priming process.

- Line 407: For consistency within the field and avoid confusion "boost-dependent" may be replaced by "activation signal-dependent".

- Line 556-557: Noteworthy, monocytes from a heterozygote CAPS patient bearing E527K/D648Y (on the same allele) were resistant to MCC950 (PMID: 38530241, Fig 5B). Therefore, contrary to the authors' claim, inhibition of the WT NLRP3 allele in heterozygote patients might not be sufficient, for an unknown reason.

- Table 5: According to the cited ref, K496 is ubiquitinated by Cbl-b only. RNF215 targets the LRR domain (not K496). In addition, recent publications on palmitoylation sites might be included as well. Especially some correspond to variant sites (C130, C898)(PMID: 39173637, PMID: 39024103, PMID: 38583156, PMID: 38949555, PMID: 38092000).

We thank the reviewers for their valuable comments which help us to improve the manuscript. We also thank the reviewers for the time and effort they dedicated to reviewing our work. We have carefully addressed each of the comments below.

Reviewers' Comments:

Reviewer #1:

Remarks to the Author:

In their clearly presented manuscript, “Mechanisms of NLRP3 activation and inhibition elucidated by functional analysis of disease-associated variants.”, Feng et al. take a methodical and comprehensive approach to characterize the functional significance of all described NLRP3 variants and study the mechanisms of specific CAPS mutations. They use recombinant HEK cells, flow cytometry, structural modeling, as well as ion changes, cold temperature, and pharmacologic approaches to produce a long awaited and clinically useful resource for clinicians and scientists.

The strengths of this paper are:

1. Logical evaluation of all reported NLRP3 variants (including common low penetrance variants and variants in other domains besides NACHT) that explores several mechanisms including cold and MCC950 responses that is relevant to CAPS patients.
2. Detailed structural modeling to visualize molecular function.
3. Authors do a nice job of not overstating and recognizing importance of clinical presentation.
4. Data and interpretation that come together for an understandable story.

We appreciate your positive feedback on our work.

The primary weaknesses

1. More discussion about the clinical CAPS spectrum as an explanation why some mutations are less straightforward.

We thank the reviewer for this suggestion. We have now included sentences below to discuss the clinical manifestation spectrum of CAPS (line 677-678 and 686-689):

“However, the same NLRP3 variant can also lead to different disease manifestations in different patients³⁶⁻⁴⁰. ” “Although our assay can distinguish between FCAS, MWS and NOMID in aggregate (**Fig. 1c**) or at a specific amino acid residue (**Fig. 1d**, **Supplementary fig. 3d**) it did not have predictive power to determine where an individual patient would fall in the clinical spectrum of CAPS.”

2. While I really like the structure figures, I am concerned that some readers not familiar with inflammasome structure and function may have difficulty interpreting them. It might be helpful to include some supplemental cartoon figures to illustrate the points they are trying to get across.

As recommended, we have now included a schematic overview describing the role of different NLRP3 variants during the structural re-arrangement that occurs when NLRP3 is activated (**Supplementary figure 6**).

Comments:

Title: Accurate and succinct”

Abstract: Well written and concise

Introduction

Concise and comprehensive and appropriate for this audience

Good review of current state of knowledge of this disease and mechanisms

Methods

Adequate detail of techniques

Results

Clear presentation of data

Figure legends adequately detailed.

Thank you for the supportive feedback.

Minor points –

The authors mention L307P (with L355P) as a common FCAS mutation but do not show any cold data.

Thank you for the suggestion. We have tested L307P along with other FCAS-associated NLRP3 variants in the cold temperature exposure assay. The results are shown in table 1 for complete data set and table 3 for specific variants which had increased activity in reponses to cold.

I don't understand vibrational entropy.

We apologize for the lack of clarity in the original manuscript. We have now included an explanation for vibrational entropy (line 552-554):

“Dynamut is an algorithm that can estimate vibrational entropy, which is a measure of protein flexibility determined by the frequencies of normal vibrations in the protein which relates to space available for molecules to move within ⁵¹. ”

Supplementary Figures

Appropriate placement of supplemental data.

Discussion

Data supports conclusions.

References – Adequate

We appreciate your positive comments.

Reviewer #2:

Remarks to the Author:

In this manuscript, Feng et al. adapted the Time of Flight for inflammasome Evaluation (TOFIE) assay in HEK293T cells and single cell analysis by flow cytometry to investigate gain-of-function of >500 natural variants of NLRP3 found in the Infevers and ClinVar databases. Although the assay has been already described by others since 2016 (PMID: 27479658), the originality of this manuscript relies on the standardized statistical analysis and the inclusion of by very far the largest numbers of NLRP3 variants in a comparative study. Results of these analysis identify gain-of-function mutations in this assay and therefore provide an important resource for CAPS diagnosis. CAPS is often efficiently treated with anti-IL1, and diagnosis is critical but pathogenicity of most NLRP3 variants is poorly characterized. Therefore, these results will certainly have immediate impact on the patients care. In addition, identification of the key residues for the control of NLRP3 activity provides also new insights on NLRP3 regulation mechanism. Nevertheless, the major weakness of this manuscript is that results rely only on one experimental system (overexpression in HEK293T) which is very artificial and highly prone to artifacts, and results are not confirmed by physiological models other than the patient symptoms for some picked mutations. As it is a rare disease with very small cohort for each mutation, and that the phenotype/genotype correlation is not strong (lots of variability between patients of the same family, some patients show symptoms from different disease entities FCAS, MWS, NOMID, which consist more of a spectrum than distinct clinical entities), the validation of the results is questionable. Therefore, although the conclusion of the manuscript may lead to major advances and impact for both the CAPS and the inflammasome fields, the manuscript requires major revisions to be accepted in Nature Immunology.

We appreciate this constructive feedback. We have now performed experiments in a myeloid cell line to confirm findings from the original manuscript. Please see additional responses to specific comments provided below:

Major points:

- In myeloid cells expressing endogenous NLRP3, NLRP3 is regulated by translational and post-translational modifications upon priming and activation signals. The TOFIE assay in HEK293T cells detects intrinsic NLRP3 gain-of-functions. Results in table 1 are based on untreated cells, therefore only constitutive active mutants can be detected, ignoring mutants that could show a lower activation threshold to priming or activation signals. In table 2, only a small selection of mutants (116/529) is investigated in response to Nigericin as an activation signal and cold temperature. The role of priming is mostly considered by the authors through the aspect of NLRP3 up-regulation and assessed by measuring NLRP3-GFP fluorescence level. Nevertheless, priming is now recognized to include complex, and very poorly understood, post-translational modifications controlling NLRP3 conformational changes, interaction with partners and subcellular relocalization (PMID: 32745339). Overexpression in HEK293T cells may bypass most of these regulations, and therefore possible gain-of-function mutants may be missed.

As suggested, we have now included results from all variants of unknown significance in NLRP3, to quantify their response to activation with nigericin or cold temperature. The data have been updated in **Table 1-3, Supplementary figure 7b and 9a**.

Regarding translational regulation of NLRP3, we have addressed this in the discussion, lines 484-498 of the revised manuscript. We surmised that some NLRP3 variants may influence this, however they were almost all found in healthy individuals, and so might act as risk factors for disease, as opposed to highly penetrant monogenic mutations. To confirm this, we tested a collection of NLRP3 variants with low baseline activity using myeloid THP-1 cells (**Supplementary fig. 4a-b**). This included Q705K as an example which had been associated with translational regulation of NLRP3, and variants that were previously classified as pathogenic or likely pathogenic variants, such as R172S, Q225P, R556X and R605G. All of these still have activity similar to WT NLRP3 in the myeloid cell line we employed. We have now described the result in the manuscript text (line 278-301):

“Furthermore, we reconstituted these likely benign variants and VUS including R172S, Q225P, R556*, R605G and Q705K into NLRP3 deficient THP-1 cells (**Supplementary fig. 4a**). We confirmed that their inflammasome activity as triggered by the NLRP3 activator, nigericin and measured by the level of IL-1 β and IL-18, could not surpass WT. We also used PMA-differentiated THP-1 cells and tested the monosodium urate (MSU) crystals, K⁺ independent NLRP3 activator, imiquimod, and another NLRP3 activator, LLOMe, which activates NLRP3 by inducing lysosomal rupture (**Supplementary fig. 4b**)⁸. Both R605G and Q705K did not have increased inflammasome activity compared to WT.”

Additionally, we agree that post-translation modification can regulate NLRP3 activities, some of which may not be reconstituted in HEK293T cells. We have now included a table summarizing all the known residues in NLRP3 that can be modified and compare these with the disease-associated NLRP3 variants (**Table 5**). Indeed, we found three NLRP3 variants that might be affected by post-translational modifications including S198N, K567E and Y861C. Therefore, we tested these variants in myeloid THP-1 cells (**Supplementary fig. 11**). These results are now included as a new paragraph in the main manuscript (line 563-603):

“Post-translational modification contributes to NLRP3 activity

Post-translational modifications (PTM) such as phosphorylation or ubiquitination can modulate NLRP3 inflammasome activity (**Table 5**)^{52,53}. Reported variants at S198, K567 and Y861 occur at sites of NLRP3 PTM, which might not be captured in the HEK293T cell line assay. Therefore, we reconstituted S198N, K567E, Y861H and Y861C into NLRP3 deficient THP-1 cells (**Supplementary fig. 11a-c**). Phosphorylation of NLRP3 at S198 by JNK1 is a key priming event that facilitates NLRP3 activation⁵⁴. Indeed, phosphorylation-resistant S198N produced less IL-1 β and IL-18 in response to nigericin when compared to similarly expressed WT NLRP3 (**Supplementary fig. 11a**). However, JNK1 is also active in HEK293T cells⁵⁵, as shown by the consistent result for S198N when compared to WT in our original screen (**Table 1**). We also saw consistently increased activity for K567E in both HEK293T cells and THP-1 cells (**Table 1, Supplementary fig. 11b**). This is in contrast to the published literature where mouse NLRP3 activity was promoted by ubiquitination at K565⁵⁶, however that result was not confirmed with human NLRP3. Additionally, phosphorylation of NLRP3 at Y861 is reported to suppress the NLRP3 activity⁵⁷. Protein tyrosine phosphatase non-receptor 22 (PTPN22) was found to dephosphorylate NLRP3 in response to inflammasome triggers, allowing efficient NLRP3 activation⁵⁷. In our original screen, variants at this residue seemed similar to WT at baseline, although with slightly enhanced responses to nigericin stimulation. Reconstituting these variants in THP-1 cells revealed that Y861C, which cannot be inhibited by phosphorylation, has increased inflammasome activity (**Supplementary fig. 11c**), while Y861H, which can still be phosphorylated, is similar to WT in the absence of an inflammasome trigger (**Supplementary fig. 11c**). Notably, the effect of Y861C was greatly diminished compared to other germline NOMID mutations such as R262P (**Supplementary fig. 11c**). Overall, these findings are consistent with the published observation that this specific variant can drive an atypical boost-dependent CAPS phenotype associated with sensorineural hearing loss rather than urticarial skin manifestations⁵⁸. This highlights the requirement for ancillary evidence when considering downgrading the classification of a specific variant based on our primary HEK293T cell screen and suggests that other complex or atypical cases may be encountered in the future requiring validation in myeloid cell lines.”

- A major limitation of the study is the use of only one read-out (ASC speck) in a very artificial system. Noteworthy, in human monocytes, NLRP3 has been shown to form alternative inflammasome without ASC speck (PMID: 27037191). Do PBMCs from CAPS patients show speck spontaneously or upon LPS priming (in conditions where pyroptosis or IL-1b are detected)? It would be informative to compare (1) mutants with high ASC50 and (2) others with low ASC50 in untreated conditions but with higher sensitivity to Nig or cold temperature, and finally (3) mutants without any detectable increased IC50 in any of the conditions, for speck formation in PBMCs ex vivo.

Given that CAPS is a rare disease, it is not feasible to collect PBMCs from the wide range CAPS patients spanning the list of variants required. Therefore, we have verified our results with a collection of NLRP3 variants (as suggested by the reviewer) in *NLRP3*^{-/-} THP-1 cells reconstituted with GFP-tagged wild-type or mutant NLRP3, and tested the release of IL-1 β and IL-18. The level of GFP-NLRP3 expression was measured using flow cytometry. These results are shown in the main figure 3 and in **Supplementary figure 3, 4, 8, 10-12**.

(1) We found that NLRP3 with high ASC50, such as R262P, R262L, R262W, G757R, L355P, K567E, T438P, T438A, E527V and E527K all had increased level of IL-1 β in response to priming and induced expression of NLRP3 with doxycycline. We also found that doxycycline treatment alone can already induce elevated level of IL-18 release. This result is consistent with our RT-PCR result that IL-18 but not IL-1 β is constitutively expressed in THP-1 cells (**Supplementary fig. 2b**). These results confirmed our findings using HEK293T cells.

(2) We also tested hypersensitive NLRP3 variants in THP-1 cells. We confirmed that two nigericin-sensitive variants, Y861H and R920Q, had increased level of cytokine release in response to nigericin when compared to WT (**Figure 3c-d, Supplementary fig. 7c-d**). For the temperature sensitive variants, we tested L413V and L355P and confirmed that they can be activated when exposed to cold (**Supplementary fig.10**).

(3) We tested a group of NLRP3 variants with unknown significant activity in our assay including R172S, Q225P, R556X, R605G and Q705K. We found that they either had a similar or reduced level of IL-1 β and IL-18 in response to nigericin or other NLRP3 stimuli such as imiquimod, LLOMe and MSU (**Supplementary fig. 4**).

Additionally, a recent study by Cosson et al. investigated a range of CAPS-associated NLRP3 variants in human PBMCs and in a myeloid cell line, U937 cells (PMID: 38530241). Their results on the activity of NLRP3 variants also verified our findings independently.

- Tables 1-3: It would be informative to add as additional columns for each mutation the associated symptom(s) (FCAS, MWS, NOMID) and whether each mutation has been identified as symptomatic when somatic or/and germline. While authors referred to patient symptoms as the main confirmation method for their data in HEK293T-based assay, it often relies only on few picked examples. Systematic analysis would be more convincing.

Thank you for the suggestion. We have now added additional columns indicating the mutation-associated symptoms and genetic information in table 1. We have now systematically analyzed the ASC50 of all pathogenic or likely pathogenic variants and their associated disease phenotype and found an increasing trend of ASC50 for variants with more severe disease phenotype (**Fig. 1c**). We also compared the germline or mosaic status of pathogenic or likely pathogenic NLRP3 variants, we showed that variants found only in mosaic patients are more active than those found in the germline (**Supplementary fig. 2c**).

- Fig 1b: Some patients have some phenotypes typical to different historical disease entities (FCAS, MWS, NOMID), and some variants are found in patients with different disease entities that correspond more to a spectrum of one disease. The authors should indicate the criteria used to link each variant to the corresponding FCAS, MWS, NOMID groups. For example, R262W is associated with FCAS and MWS in the Infefers database. What are the criteria to associate R262W to FCAS rather than MWS in Fig 1b? In addition, to be more convincing, all variants for one given site should be included in the figure. For example, R262Q could be added. Additional examples, other than R262, A441 and 629 could also be added in fig 1b or as supplemental figures.

Thank you for pointing this out. We have now updated the figure and text to clarify that R262W and other variants may be associated with different parts of the CAPS spectrum (**Fig. 1d**). The disease-associated phenotypes for NLRP3 variants have been included in table 1 where available. We also compared the ASC50 for all variants any one given site when they are associated with a different disease phenotype (**Supplementary fig. 2d-e**).

- A subset of 116 variants among the 529 variants included in the study has been investigated in response to nigericin treatment. The criteria to choose these 116 variants are unclear as many “unknown significance” variants (table 1 column E) were not included in table 2. For the analysis to be comprehensive, all variants without constitutive activity detected in table 1 may be tested in response to nigericin signal. In addition, other

activation signals could be tested including particles and K⁺-independent stimuli (Imiquimod-derivatives), as mechanisms and regulatory structure may differ (as does nigericin- and temperature-dependent activation mechanism)

Our initial analysis came from the variants of unknown significance present for NLRP3 in the INFEVERS database, however we have now included many such additional variants from ClinVar in response to nigericin and cold temperature treatment as well (**Table 1**, **Supplementary fig. 7b, 9a**).

Given that HEK293T cells are not good phagocytes and NLRP3 cannot be efficiently activated by crystals or Imiquimod-derivatives, we employed myeloid THP-1 cells for this purpose. We reconstituted these cells with two NLRP3 variants of unknown significance which appear inactive in HEK293T cells, R605G and Q705K. We then stimulated the PMA-differentiated THP-1 cells with monosodium urate (MSU) crystals, a K⁺ independent NLRP3 activator imiquimod, or another NLRP3 activator, LLOMe, which activates NLRP3 by inducing lysosomal rupture (**Supplementary fig. 4b**). Both NLRP3 variants R605G and Q705K did not exhibit an increased level of IL-1 β or IL-18 secretion when compared to WT NLRP3 (**Supplementary fig. 4b**).

- In an attempt to validate the experimental system, the authors should compare the expression level of NLRP3 and ASC in the HEK293T vs primary myeloid cells of, at least, some human cell lines.

Thank you for the suggestion. We have now compared the expression level of NLRP3 and ASC in HEK293T cells, myeloid cell lines (THP-1 and U937), and in human PBMCs (**Supplementary fig. 2c**). We showed that NLRP3 and ASC are expressed in THP-1, U937 and in human PBMCs but not in HEK293T cells (**Supplementary fig. 2c**).

- Line 247-253: NEK7 has been shown to be dispensable in several human cells for inflammasome activation. Does HEK293T cells used in this study express NEK7 and is it required for ASC oligomerization in the ASC50 assay? Use of NEK7 KO HEK293T would definitely rule out the impact of NEK7-NLRP3 interaction in the activity of the LRR mutants.

Yes, HEK293T cells express endogenous NEK7 (**Supplementary fig. 2c**). As requested, we knocked out NEK7 in THP-1 cells (**Supplementary fig. 8**). This confirmed that NEK7 can contribute to NLRP3 activity and the hyperactivity of G757R in THP-1 cells (**Supplementary fig. 8a-d**). However, Y861H and R920Q remained hyper-responsive to nigericin stimulation in THP-1 cells even in the absence of NEK7, suggesting that NEK7

does not contribute to the increased sensitivity of Y861H and R920Q to nigericin stimulation (**Supplementary fig. 8e-h**).

- Is temperature-induced activity in HEK293T cells a common feature of all FCAS mutations, and a discriminant parameter between the FCAS vs MWS or NOMID. To correlate FCAS and cold-induced activation in HEK293T cells, it may be informative to distinguish FACS and CAPS undefined mutations in table 2 and in Fig4A. In addition, it would be informative to test cold-induced activation in HEK293T cells for all well characterized MWS and NOMID, as it has been done with FCAS.

Thank you for the suggestion. We have now combined the original table 1 with table 2, and included the disease phenotype in an extra column (**Table 1**). In the new table 3, we have included variants with increased activity in response to cold, and found that many of them are associated with FCAS.

As suggested, we have also now tested all NLRP3 variants associated with FCAS under low temperature (**Table 1**). Indeed, we found that all FCAS-associated hyperactive NLRP3 variants had increased ASC50 in response to cold (**Table 3**). The overall design of our screening approach was to maximize the signal for NLRP3 activating variants at baseline, which means that our window to see increased sensitivity to cold temperature on top of this is smaller, especially for highly active MWS and NOMID variants. Nevertheless, we also tested a few MWS and NOMID variants and found that some of these are also slightly more active at a lower temperature (**Table 1 and 3**). Future studies would be required in order to determine any potential association or physiological relevance for these observations.

Minor points

- Line 54 : “ATP hydrolysis which is important to induce structural rearrangement and inflammasome assembly”. This sentence is confusing and should be modified to clearly indicate that ATP hydrolysis is important to switch from the active to the inactive form and not the reverse (PMID: 35114687).

Yes, we have updated the manuscript text and included the reference as the reviewer suggested (line 91-93):

“The NACHT domain of NLRP3 mediates ATP hydrolysis which is important for switching the conformation of NLRP3 from the active to the inactive form³⁻⁵.”

- Line57: transcription-independent priming has also been described (PMID: 22948162)

Yes, we have updated the manuscript text and included the reference as the reviewer suggested (line 106-109):

“Priming usually involves NFκB signaling downstream of Toll-like receptors (TLRs) or another signal which leads to increased protein synthesis of NLRP3 and pro-interleukin (IL)-1β and also deubiquitination of NLRP3^{2,6}.”

- Line 97: the reference PMID: 36562561 is missing, In addition a preprint showed recently that primary monocytes from patients bearing NLRP3 E525K are resistant to MCC950 and could be cited as well (doi:<https://doi.org/10.1101/2023.09.22.558949>)

Yes, we have updated the manuscript text and included the reference as the reviewer suggested (line 154-156):

“However, MCC950 failed to inhibit inflammasome activation in macrophages derived from transgenic mice carrying the L355P NLRP3 variant ³¹, and in human monocytes simultaneously expressing E527K and D648Y variants ³⁰.”

- Supplementary fig 1C: this is not very clear how the middle and the right panels relate one to another. It may help to indicate the ASC50% in the middle panel. In addition, the choice of identical color (blue and red) for different things (Speck-/speck+ vs mutant/WT) is also confusing.

Thank you for the suggestion. We have now updated the color in the graph and included the ASC50 calculation in the middle panel (**Supplementary fig. 1f**).

- Line 142: to increase clarity, the authors may indicate that positive ASC50 corresponds to NLRP3 hyperactivity. Apparently, these mutants are distributed all across the NACHT domain and not concentrated in the NBD and HD2 subdomains as stated by the authors. Related to this last point, the authors may indicate how this “concentration” was assessed (method to calculate the frequency values?).

We have now updated the manuscript text (line 203-205):

“We defined ASC50 as the relative difference of EC50 between the variant and WT, and positive ASC50 corresponds to NLRP3 hyperactivity.”

We also included a graph showing the percentage of NLRP3 mutations identified in a specific domain relative to the domain length (**Fig. 1b**). We showed that the NBD and HD2 had more highly active NLRP3 mutations compared to other domains.

- table 1: The authors should indicate what means “out of scale”.

Thank you for the suggestion. We have included the following text in table 1 to explain “out of scale”:

“Out of scale indicates ASC50 value is much lower than WT and thus cannot be accurately calculated.”

- Table 2: It would be helpful if the author can provide a cutoff to identify mutants hypersensitive to nigericin activation signal. For example, authors conclude that Q705K (Nig ASC50=0.116) is not pathogenic, but that R920Q, Y861H, Y861C are hypersensitive to Nig (Nig ASC50=0.128, 0.146, 0.169 respectively), while Nig ASC50 are very similar.

We have now grouped all NLRP3 variants with increased activity in response to nigericin in the new table 2. Instead of a cut-off value, we provided two-tailed t-test analysis for these variants, so their results can be better interpreted (**Table 2**).

- tables1-3: It may help to add the NLRP3 domain corresponding to each mutation. In addition, the same order should be used for different substitutions of one aa (Y861H and Y861C are inverted between table 1 and 2)

Thank you for the suggestion. We have now included an extra column in table 1-3 to show the domain for each variant.

- It would be also useful to add in table 2, the data obtained in untreated condition (table1) to compare more easily the results of the different experimental conditions for one given variant on one single sheet. In addition, graphical representation of the data would be useful (using Fig1a as a model)

Yes, we have updated the table as suggested. The result for NLRP3 variants under different conditions are all included in the same table now (**Table 1**). Instead of graphical representation of these data, we summarized NLRP3 variants with increased activity in response to nigericin or cold, or resistant to MCC950 in separate tables to present the result in a more informative way than it was previously (**Table 2-4**).

- This is confusing that the authors use in this manuscript a different nomenclature than in the Infevers database to label the mutations (resulting in a 2 aa shift). In addition, in Line 393-396 the Infevers nomenclature is used, inconsistently with the rest of the manuscript.

Thank you for pointing this out. We have updated the nomenclature in the text as suggested. The Infevers database used variant names as first published or as submitted, and these were mostly for the 1034 aa version (isoform e) and a few for the 1036 aa version (isoform a). Isoform a is used in several databases such as Gnomad, and the NLRP3 plasmid we used for reconstitution in this manuscript encodes the 1036aa, longer version of NLRP3. Therefore, we try to use the 1036 aa NLRP3 nomenclature throughout, but for readers who may be accustomed to the shorter 1034 aa NLRP3 nomenclature, we have now included both in table 1-4.

- Line 235: "KCI did inhibit nigericin-induced activation for all". According to table2, not all mutants with increased sensitivity to Nig were tested in presence of Nig and increased KCI. What are the criteria used to pick the 5 mutants tested for sensitivity to KCI?

We apologized for the confusion. We have now included a table ranking all the nigericin sensitive variants based on statistical significance (**Table 2**). We simplified the result to just show Y861H and R920Q as these two are the main examples discussed in this manuscript.

- Line 242: this conclusion is very similar to the conclusion of a recent preprint (Cosson et al. ref 51), which may be cited here.

Thank you for the suggestion. We have now cited the work from Cosson et al where appropriate.

- Most mutations symptomatic when somatic are not found as germline, suggesting that mutations with strong gain of functions may be lethal as germline (PMID: 31816408). Therefore, symptomatic somatic mutations may correspond to mutation with strong gain-of-function. Correlation between ASC50 score and germline vs somatic symptomatic mutations may be informative.

Thank you for this insight. We analyzed the correlation between ASC50 score and germline/somatic variants as suggested (**Supplementary fig. 2c**). Indeed, we found that variants found only in mosaic patients are more active than those found in the germline.

- The manuscript relies exclusively on highly processed data. Additional FACS plots (for each NLRP3-GFP increment gateings), for WT and few mutants would be informative for the reader. In addition, microscopy images of FACS sorted Speck+ and Speck- cells to control the ASC specks are necessary.

Thank you for the suggestion. As an example, we showed the NLRP3-GFP increment gating in supplementary figure 1d. As suggested, we also showed microscopy images for speck+ and speck- cells using imaging flow cytometry (**Supplementary fig. b-c, e**).

- Fig5a,b : some mutations are activated by MCC950 treatment (for example at 12h, a dot with MCC950(ASC50) around 0.42 and UT(ASC50) around 0.1). It is surprising that MCC950 may be able to bind the active conformation of NLRP3. The authors should comment on that and confirm their data in myeloid cells competent of NLRP3 activation.

Thank you for pointing this out. We further investigated this variant, G461C, that stands out from the initial screen for 12h incubation with MCC950. However, this is not observed at 4h MCC950 treatment, suggesting that the result for G461C at 12h is an outlier. Indeed, we confirmed that the ASC curve failed to be generated in one of the repeats and thus leads to a false value which has now been removed from analysis (figure shown below). Other variants with a similar issue have now been reanalyzed. Additionally, we have now combined the result for MCC950 treatment into table 1 for direct comparison for baseline NLRP3 activity.

- Line380-383: It should be added here that in addition to the upregulation of NLRP3 expression, priming results in NLRP3 post-translational modifications and relocalization that render NLRP3 responsive to activation signal (PMID: 32745339).

Yes, we have now updated the manuscript text as suggested (line 705-707):

“Additionally, priming also results in NLRP3-post-translational modifications and relocalization that can alter NLRP3 responsiveness to an activation signal.”

- Line 382-383: to validate this hypothesis, authors should test whether (1) variants activated by nigericin (table2), and (2) variants hyper-responsive to signal 2 in the cited preprint (ref 51) show higher NLRP3 expression level.

We now included a correlation analysis for the ASC50 level and the expression level of GFP-NLRP3 for both Infevers and ClinVar variants (**Supplementary fig. 2a**). Our results showed that the NLRP3 activity assessed by ASC50 is not associated with the NLRP3-GFP expression level.

- Line 393-396: The gain-of-function of Q705K is highly discussed including in functional assay (PMID: 22529966), and although it is a frequent variant in the healthy individuals, its prevalence is increased in autoinflammatory patients (PMID: 32477355). This suggest that this mutant is a gain-of-function pathogenic mutant although it may not be sufficient to trigger the disease depending on the environmental exposome and other genetic factors.

Our data agree that Q705K is a variant that might not trigger disease in isolation, but may do so in response to environmental stimuli or other genetic factors. We discuss this as a variant that may be associated with autoinflammatory disease as suggested, although as a risk-factor, and not as a penetrant, pathogenic NLRP3 mutation.

- Different fixation methods (methanol or PFA) are used depending on the cell treatments. The authors should explain why different protocol were used and provide controls to assess the stability of ASC50 values following different fixation procedures. Can the values in the different tables be compared?

To investigate the activity of NLRP3 variants in response to cold temperature exposure, we incubated transfected HEK293T cells in either 37 or 32 degrees. To minimize the disruption by temperature changes that could trigger activation of some NLRP3 variants, we used pre-warmed 4% PFA instead of pre-chilled methanol in this experiment.

We compared the ASC50 for control variants used in each plate between two different fixation procedures (figure shown below) and found no significant difference between them. Therefore, we have now combined table 1 and 2. Due to space limitations, we did not include this figure in the main manuscript.

Thank you for the helpful comments. We have now revised the manuscript as suggested.

Reviewer #2 (Remarks to the Author):

The authors replied adequately to most points. Their effort to validate some of the results in myeloid cell lines is highly appreciated. The last remaining point concerns the description of the statistical analysis, that I think should be more detailed to ease the understanding by the reader (see below). I also raised few minor points to be edited, to better reflect the literature.

Once the authors would have addressed these points, the manuscript would be suitable for publication in Nature Immunology. This comprehensive analysis of all NLRP3 natural variants will be a highly valuable resource for CAPS diagnosis and the understanding of NLRP3 activation and regulation mechanisms.

Major points:

- Statistical analysis. To my opinion, the statistical criteria to determine which mutant may be auto-activated, hypersensitive to Nig or activated by low temp, are still unclear. The method to calculate p values should be added to the methods section. In tables 2 and 3, it would be helpful to add which values are compared in the headline of the p-value column (as done in table 4). This is unclear whether the compared values are: (1) Nig Vs UT for a given variant or (2) Nig.variant vs Nig.WT. Adding p-values for UT ASC50 for each given variant compared to WT NLRP3 in table 1 should be informative as well. I also could not find the number of independent repeats.

We have now added method used for p value calculation to clarify the statistical criteria. We also revised the table to include number of repeats and p-values for variants in table 1-4, and headlines of the column for clarity.

Minor points:

- Line 71: Priming controls NLRP3 not only by its deubiquitination, but also many other modifications including -but not limited to- JNK1-dependent phosphorylation (PMID: 28943315). This line may be modified to reflect all the complexity of the NLRP3 priming process.

We have now revised the sentence to reflect the complexity of NLRP3 priming process. (Line 68-72):

“Priming usually involves NF κ B signaling downstream of Toll-like receptors (TLRs) or another signal which leads to increased protein synthesis of NLRP3 and pro-interleukin (IL)-1 β and post-translational modifications of NLRP3 such as deubiquitination, phosphorylation, sumoylation and palmitoylation (PMID: 36336552).”

- Line 407: For consistency within the field and avoid confusion “boost-dependent” may be replaced by “activation signal-dependent”.

We have now revised the term as indicated.

- Line 556-557: Noteworthy, monocytes from a heterozygote CAPS patient bearing E527K/D648Y (on the same allele) were resistant to MCC950 (PMID: 38530241, Fig 5B). Therefore, contrary to the authors’ claim, inhibition of the WT NLRP3 allele in heterozygote patients might not be sufficient, for an unknown reason.

We agree that inhibition of the WT NLRP3 allele may not be sufficient. CAPS patients carrying drug-resistant variants should be carefully considered for therapies targeting NLRP3.

- Table 5: According to the cited ref, K496 is ubiquitinated by Cbl-b only. RNF215 targets the LRR domain (not K496). In addition, recent publications on palmitoylation sites might be included as well. Especially some correspond to variant sites (C130, C898)(PMID: 39173637, PMID: 39024103, PMID: 38583156, PMID: 38949555, PMID: 38092000).

We have now revised table 5 to include recent publications on palmitoylation.